# VRAda: A Variance Reduced Adaptive Algorithm for Stochastic Parameter-Agnostic Minimax Optimizations

## Abstract

Stochastic parameter-agnostic minimax optimization provides a novel avenue for adjusting learning rates without relying on problem-dependent parameters, bridging the gap between theoretical and empirical machine learning results. While previous studies have successfully decoupled the timescales of primal and dual variables and proposed unified parameter-agnostic algorithms for minimax optimizations, the problem of varying inherent variances within the stochastic setting persists. Such variance degradation affects the desired ratio of learning rates. Intuitively, variance-reduced techniques hold the potential to address this issue efficiently. However, they require manually tuning problem-dependent parameters to attain an optimal solution. In this paper, we introduce the Variance-Reduced Adaptive algorithm (VRAda), a solution addressing varying inherent variances and enabling the parameter-agnostic manner in stochastic minimax optimizations. Theoretical results show that VRAda achieves an optimal sample complexity of $O(1/\epsilon^3)$ without large data batches, enabling it to find an $\epsilon$-stationary point on non-convex-strongly-concave and non-convex-Polyak-Łojasiewicz objectives. To the best of our knowledge, VRAda is the first variance-reduced adaptive algorithm designed specifically for parameter-agnostic minimax optimization. Extensive experiments conducted across diverse applications validate the effectiveness of VRAda.

## 1 Introduction

In this paper, we consider the following stochastic minimax optimization problem:

$$\min_{x \in \mathbb{R}^{d_1}} \max_{y \in \mathcal{Y}} f(x, y) = \mathbb{E}_{\xi \in \mathcal{D}}[F(x, y, \xi)], \tag{1}$$

where $\mathcal{D}$ is a dataset with unknown data distribution, from which we can draw i.i.d. samples, $\mathcal{Y} \subset \mathbb{R}^{d_2}$ is closed and convex, and $f : \mathbb{R}^{d_1} \times \mathbb{R}^{d_2} \to \mathbb{R}$ is non-convex in $x$. We call $x$ the primal variable and $y$ the dual variable. In fact, problem in equation 1 is widely used in many machine learning applications, such as adversarial training Goodfellow et al. (2014b); Miller et al. (2020), Generative Adversarial Network (GAN) Arjovsky et al. (2017); Goodfellow et al. (2014a), deep Area Under the Curve (AUC) Yuan et al. (2021; 2022), reinforcement learning Dai et al. (2017); Modi et al. (2021), and sharpness-aware minimization Foret et al. (2020); Qu et al. (2022).

While numerous studies Nouiehed et al. (2019); Lin et al. (2020); Lu et al. (2020) have devised efficient algorithms to tackle the minimax problem in equation 1, they require knowledge of problem-dependent parameters, e.g., smoothness parameter $L$, bounded gradient $G$, and strong concavity $\mu$. However, acquiring these problem-dependent parameters is very difficult in realistic machine learning applications, leading to a significant gap between theory and empirical performance and limiting the practical utility of these algorithms. To bridge this gap, adopting a parameter-agnostic approach has proven effective Ward et al. (2020); Xie et al. (2020); Antonakopoulos et al. (2020). Parameter-agnostic algorithms enable automatic adjustment of learning rates without prior knowledge of problem-dependent parameters, relying solely on cumulative gradient information to achieve the desired convergence ratio. Some studies have designed corresponding algorithms on non-convex optimizations by adaptive optimizers, like AdaGrad Duchi et al. (2011), AMSGrad Reddi et al. (2018), and STORM+ Levy et al. (2021). These approaches are appealing due to their robust handling of hyper-parameter selection and their ability to achieve rapid empirical convergence.

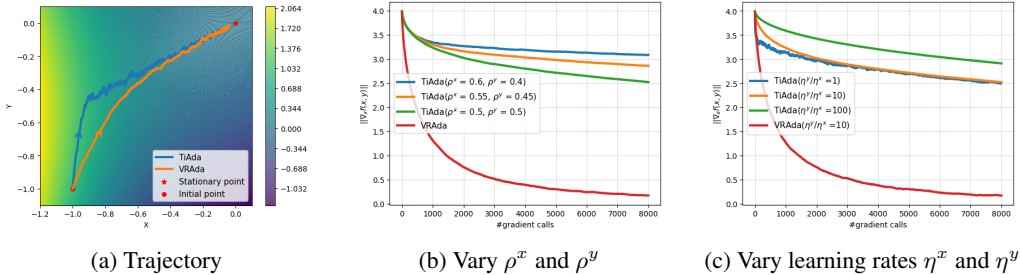

(a) Trajectory       (b) Vary $\rho^x$ and $\rho^y$       (c) Vary learning rates $\eta^x$ and $\eta^y$

Figure 1: Comparison between VRAda and TiAda on the test function $f(x,y) = -x^3 + xy - y^2$, with more noise on $y$ than $x$. Here, $\rho^x$ and $\rho^y$ are the forces with which the scales of $x$ and $y$ are separated, $\eta^x$ and $\eta^y$ are the initial learning rates in Li et al. (2022). Figure 1a shows the trajectory of the two algorithms and the background color demonstrates the function value $f(x,y)$. In Figure 1b, although varying the values $\rho^x$ and $\rho^y$ in TiAda, VRAda still outperforms TiAda continuously. Figure 1c illustrates the convergence results varying the learning rates of TiAda.

However, a recent study TiAda Li et al. (2022) has highlighted that the direct extension of adaptive optimizers to minimax optimizations may not guarantee convergence without prior knowledge of problem-dependent parameters. Additionally, TiAda develops an efficient time-scale separation algorithm to fit the parameter-agnostic manner. Although they have obtained significant achievement in the deterministic setting, they dismiss the varying inherent variances, coming from $\nabla_x f(x, y, \xi^x)$ and $\nabla_y f(x, y, \xi^y)$[1], in the stochastic setting. As a result, this issue may degrade the capability of automatically adjusting to the desired ratio of learning rates. Therefore, reducing the two inherent variances should be helpful for stochastic parameter-agnostic minimax optimizations.

Recently, variance-reduced techniques have demonstrated their effectiveness in handling noisy gradients Fang et al. (2018); Zhou et al. (2020); Nguyen et al. (2021). These techniques utilize well-designed estimators for updates, replacing the sole reliance on stochastic gradients. While showing improved convergence rates in non-convex optimization Allen-Zhu & Hazan (2016), they often require specific conditions, such as large batch sizes or fine-tuning problem-dependent parameters to achieve optimal solutions. Moreover, when addressing minimax optimizations, it becomes crucial not only to mitigate varying inherent variances without resorting to large batches or anchor points but also to automatically adjust the learning rate ratio on the separating two-time scales.

In this paper, we introduce an effective parameter-agnostic algorithm named Variance-Reduced Adaptive (VRAda). To handle inherent variances, VRAda rapidly detects and responds to errors in both variables $x$ and $y$. When VRAda detects a substantial error in either variable, it reduces the update rate further based on historical gradient estimations. Regarding time-scale separation, VRAda regulates the update of $x$ based on the current state of $y$. Specifically, if $y$ has not yet reached the current optimum, VRAda will slow down the update of $x$ in response to $y$'s update situation. Once $y$ approaches optimality, $x$ will update at its original speed, achieving adaptive time-scale separation.

We use the non-convex-strongly-concave function $f(x,y) = -x^3 + xy - y^2$ as a toy example to assess the efficacy of VRAda. As depicted in Figure 1a, the trajectory of TiAda Li et al. (2022) exhibits noticeable divergence compared to the optimal path, especially on $y$-axis. This divergence may be attributed to TiAda's limited ability to autonomously adjust learning rates to counteract the adverse effects of substantial inherent variance on $y$, particularly during the initial epochs. Figures 1b-1c demonstrate that VRAda consistently outperforms TiAda across all scenarios. Furthermore, TiAda requires meticulous parameter selection for time-scale enforcement and initial learning rates; otherwise, it struggles to achieve convergence. This observation underscores the robustness of VRAda. In summary, the main contributions of this paper can be summarized as follows:

- We introduce VRAda, a fully parameter-agnostic variance-reduced adaptive algorithm designed for stochastic Non-Convex-Strongly-Concave (NC-SC) and Non-Convex-Polyak-Łojasiewicz minimax (NC-PL) optimization settings. VRAda effectively reduces the varying inherent variances of both variables after necessary time-scale separation without the need

---

[1]Note that even when $\xi^x$ and $\xi^y$ originate from the same dataset, i.e., $\xi^x, \xi^y \in \mathcal{D}$, distinct sub-problems concerning variables $x$ and $y$ still entail varying inherent variances during gradient calculations.

for large batch sizes or any prior knowledge of problem-dependent parameters. It achieves the identification of an $\epsilon$-stationary point with an optimal complexity of $O(1/\epsilon^3)$ in both settings, outperforming the parameter-agnostic algorithm TiAda Li et al. (2022) and aligning with existing parametric minimax algorithms.

- We conduct our proposed VRAda algorithm on several learning tasks and datasets compared with TiAda by varying its initial learning rates and ratios. (1) Four different test functions for showing the robustness of VRAda on different minimax optimization problems, (2) deep AUC Yuan et al. (2021) with an NC-SC objective, and (3) training Wasserstein-GANs Arjovsky et al. (2017) to validate the NC-PL objective. In all tasks, VRAda performs consistently better than TiAda or TiAda-Adam.

## 2 RELATED WORK

**Stochastic Minimax Optimizations.** Stochastic minimax optimization has gained significant traction in various machine learning applications, like adversarial training, deep AUC, GANs, and policy evaluation. The predominant algorithms for addressing this challenge are rooted in stochastic gradient descent ascent Nouiehed et al. (2019); Lin et al. (2020); Lu et al. (2020); Yan et al. (2020). These algorithms typically involve one primal variable update followed by one or multiple steps of dual variable updates. Notably, they can achieve a sample complexity of $O(\epsilon^{-4})$ in stochastic settings Nouiehed et al. (2019); Lin et al. (2020); Yang et al. (2020). Later on, some accelerated algorithms based on adaptive learning rates have been extended to minimax optimization, both in theory and practice, SC-SC Antonakopoulos et al. (2020), NC-SC Yang et al. (2022a); He et al. (2022); Huang et al. (2023), NC-PL Huang (2023); Guo et al. (2023). For example, Huang et al. (2023) designs a fast AdaGDA method based on the momentum technique, and Guo et al. (2023) proposes PES by concerning the primal objective gap and the duality gap under the NC-PL setting.

**Parameter-Agnostic Algorithms.** The parameter-agnostic manner requires the algorithms to automatically adjust the hyper-parameters, such as learning rate, without relying on any problem-dependent parameters to achieve convergence. Some adaptive optimizers obtain this property Duchi et al. (2011); Reddi et al. (2016); Levy et al. (2021). However, they only focused on non-convex optimizations, which cannot be directly used on minimax optimizations. Later on, TiAda Li et al. (2022) extends their ability to minimax optimizations by separating the two-time scales. In addition, parameter-agnostic algorithms have been widely developed in online learning, e.g., Beygelzimer et al. (2015) for online boosting, Xu et al. (2020a) for online reinforcement learning with Gaussian processes, and Hanneke et al. (2023) for multi-class online learning, where the goal of the learner is to compete with the performance of the best function $f$ to achieve small regret.

**Variance-Reduced Techniques.** Variance-reduced techniques have gained prominence in stochastic optimization, enhancing algorithm efficiency in the presence of noise. Notable approaches include SVRG Johnson & Zhang (2013); Reddi et al. (2016), SPIDER Fang et al. (2018), and STORM Cutkosky & Orabona (2019); Levy et al. (2021), contributing to the acceleration of stochastic optimization algorithms. SPIDER leads to fast HAPG Shen et al. (2019) and SRVR-PG Xu et al. (2020b). Momentum-based techniques like ProxHSPGA Pham et al. (2020) and IS-MBPG Huang et al. (2020) arise from STORM's principles. More recently, Zhang et al. (2021) introduces Truncated Stochastic Incremental Variance-Reduced Gradient (TSIVR-PG) to mitigate unverifiable importance weight assumptions, establishing global convergence, even with policy overparameterization.

## 3 THE PROPOSED ALGORITHM

### 3.1 DESIGN CHALLENGES

Although TiAda Li et al. (2022) has been demonstrated a parameter-agnostic algorithm capable of automatically adjusting learning rates to the desired ratio, eliminating the problem-dependent parameters like smoothness parameter $L$, gradient bound $G$, and strong concavity parameter $\mu$. However, a persistent challenge in the stochastic parameter-agnostic minimax setting is the unresolved issue of varying inherent variances of the primal variable $x$ and the dual variable $y$. When these inherent variances have obvious differences, the optimizer faces difficulties adapting to the optimal trajectory, leading to deteriorating convergence rates. In addition, the presence of inherent variance in

stochastic gradients can negate the theoretical advantages of momentum terms Yuan et al. (2016). Therefore, the selection of learning rate decay values for the momentum decay parameter must be customized to accommodate the varying inherent variances associated with $x$ and $y$.

Variance-reduced techniques Johnson & Zhang (2013); Reddi et al. (2016), have gained recognition due to their commendable convergence analysis and optimal theoretical results. They excel in obtaining more accurate gradient values by employing specific estimators tailored for stochastic settings. However, these variance-reduced algorithms mainly focus on theoretical aspects, where they need to manage anchor points alongside judicious choices of large batch sizes or manual adjustments to problem-dependent parameters, posing additional hurdles.

In summary, our objective is to devise an efficient algorithm capable of identifying an $\epsilon$-stationary point for the minimax optimization problem in equation 1. This endeavor addresses three key challenges: (1) eliminating the necessity for large batches or anchor points; (2) adaptively reducing the inherent variance for both the two sub-problems, pertaining to variables $x$ and $y$; (3) achieving the fully parameter-agnostic manner without any problem-dependent parameters. To tackle these challenges, we introduce a corresponding algorithm called Variance-Reduced Adaptive (VRAda), detailed in Algorithm 1, which we will comprehensively present in the following subsection.

---

**Algorithm 1** Learning procedure of AVRAM method.

---

**Initialization:** $(x_1, y_1), 0 < \gamma < \lambda$;

 1: **for** $t = 1$ to $T$ **do**
 2:     sample $\xi_t^x$ and $\xi_t^y$;
 3:     **if** $t = 1$ **then**
 4:         $v_t = \nabla_x f(x_t, y_t; \xi_t^x), w_t = \nabla_y f(x_t, y_t; \xi_t^y)$;
 5:     **else**
 6:         Update the estimators $v_t$ and $w_t$ via equation 2;
 7:     **end if**
 8:     Update the momentum $\beta_{t+1}$ via equation 3;
 9:     Update $\alpha_t^x$ and $\alpha_t^y$ via equation 6;
10:     Update learning rates $\eta_t^x$ and $\eta_t^y$ via equation 7;
11:     $x_{t+1} = x_t - \eta_t^x v_t, \ \ y_{t+1} = y_t + \eta_t^y w_t$
12: **end for**

---

## 3.2 Algorithm Description

Some variance-reduction techniques Johnson & Zhang (2013); Reddi et al. (2016); Fang et al. (2018) need either the maintenance of an anchor point or the utilization of large data batches in each epoch, resulting in high sample complexity. STORM Cutkosky & Orabona (2019); Levy et al. (2021) breaks new ground by achieving the nearly optimal results while sidestepping the aforementioned requirements, which can address the first challenge. By implementing a refined momentum-based gradient update, implicit variance reduction is achieved for both variables. We modify the estimator of STORM to our focused minimax optimization as follows:

$$
\begin{aligned}
v_t &= \nabla_x f(x_t, y_t; \xi_t^x) + (1 - \beta_t)(v_{t-1} - \nabla_x f(x_{t-1}, y_{t-1}; \xi_t^x)), \\
w_t &= \nabla_y f(x_t, y_t; \xi_t^y) + (1 - \beta_t)(w_{t-1} - \nabla_y f(x_{t-1}, y_{t-1}; \xi_t^y)).
\end{aligned}
\tag{2}
$$

Though STORM has achieved success in many scenarios Kavis et al. (2022); Jiang et al. (2022); Liu et al. (2023), they still need to manually adjust the problem-dependent parameters to achieve optimal results. More specifically, the momentum parameter $\beta_t$ in original STORM Cutkosky & Orabona (2019) is $\beta_t = c\eta_{t-1}^2$, where $c$ depends on $L, G, \sigma$. As a result, STORM cannot be directly in the focused parameter-agnostic objective, and we need new ways to build $\beta_t$ to tackle the second challenge. Inspired by Levy et al. (2021), to adaptively reduce the inherent variance of $x$ and $y$ respectively, we give the definition of $\beta_t$ as follows:

$$
\beta_{t+1} = \frac{1}{\left(1 + \sum_{i=1}^{t} \max\{\|\nabla_x f(x_i, y_i; \xi_i^x)\|^2, \nabla_y f(x_i, y_i; \xi_i^y)\|^2\}\right)^{2/3}}.
\tag{3}
$$

Note that Levy et al. (2021) is only designed for stochastic non-convex optimization problems, and it leverages the historical information of $x$ to calculate $\beta_t$. When using different $\beta_t$ values for variables

$x$ and $y$, denoted as $\beta_t^x$ and $\beta_t^y$, managing variance becomes challenging due to the varying variances originating from gradient calculations of the two sub-problems: $\nabla_x f(x, y, \xi^x)$ and $\nabla_y f(x, y, \xi^y)$. This makes it difficult to maintain consensus agreement in minimax optimizations and achieve the optimal solution, potentially causing slower updates for $x$ compared to $y$ Lin et al. (2020); Yang et al. (2022b). As such, it cannot be directly used in our focused problem, and hence we will introduce the reason why we update $\beta_t$ based on equation 3 in detail.

For notational convenience in the proofs, we denote $\epsilon_t^x := v_t - \nabla_x f(x_t, y_t)$, $\epsilon_t^y := w_t - \nabla_y f(x_t, y_t)$. When getting the stochastic gradient on each epoch $t$, both the primal variable $x$ and the dual variable $y$ will create some variance compared to the estimators $v_t$ and $w_t$. Specifically, based on the update rule of $v_t$ and $w_t$, we have the following error dynamics:

$$\epsilon_t^x = (1 - \beta_t)\epsilon_{t-1}^x + \beta_t(\nabla_x f(x_t, y_t; \xi_t^x) - \nabla_x f(x_t, y_t)) + (1 - \beta_t)Z_t^x,$$
$$\epsilon_t^y = (1 - \beta_t)\epsilon_{t-1}^y + \beta_t(\nabla_y f(x_t, y_t; \xi_t^y) - \nabla_y f(x_t, y_t)) + (1 - \beta_t)Z_t^y, \tag{4}$$

where $Z_t^x = (\nabla_x f(x_t, y_t; \xi_t^x) - \nabla_x f(x_{t-1}, y_{t-1}; \xi_t^x)) - (\nabla_x f(x_t, y_t) - \nabla_x f(x_{t-1}, y_{t-1}))$, $Z_t^y = (\nabla_y f(x_t, y_t; \xi_t^y) - \nabla_y f(x_{t-1}, y_{t-1}; \xi_t^y)) - (\nabla_y f(x_t, y_t) - \nabla_y f(x_{t-1}, y_{t-1}))$. In equation 4, we can find that the momentum parameter $\beta_t$ is the key factor to control both variance term $\nabla_x f(x_t, y_t; \xi_t^x) - \nabla_x f(x_t, y_t)$ and $\nabla_y f(x_t, y_t; \xi_t^y) - \nabla_y f(x_t, y_t)$. To avoid reliance on problem-dependent parameters and account for the varying inherent variances, we adopt a strategy that considers the historical gradient information for both variables and selects the maximum gradient Euclidean norm. Essentially, $\beta_t$ serves as a filter in this context, influenced by the stochastic gradients of $x$ and $y$. When one of the variables exhibits a substantial error and generates a larger stochastic gradient, $\beta_t$ responds by quickly decreasing its own value. Conversely, when one of the variables has a larger error but produces a smaller stochastic gradient, $\beta_t$ filters this information and adjusts its value based on another relatively reasonable stochastic gradient value. This adaptive mechanism is crucial for our convergence guarantee. Since cumulative historical gradient information is incremental, $\beta_t$ naturally decreases as the number of epochs increases, resulting in reduced variance.

The key point to address the last challenge is how to separate the time scales of $x$ and $y$. Based on our analysis, $Z_t^x$ and $Z_t^y$ can be upper-bounded by $\|Z_t^x\|^2 \le 8L^2((\gamma\eta_{t-1}^x)^2\|v_{t-1}\|^2 + (\lambda\eta_{t-1}^y)^2\|w_{t-1}\|^2)$ and $\|Z_t^y\|^2 \le 8L^2((\gamma\eta_{t-1}^x)^2\|v_{t-1}\|^2 + (\lambda\eta_{t-1}^y)^2\|w_{t-1}\|^2)$. It can be observed that they are very related to the learning rates due to the smooth property. Therefore, it is necessary to properly select learning rates to eliminate the impact of the smooth parameter $L$. Note that the learning rate $\eta_t$ in STORM Cutkosky & Orabona (2019) is defined as follows:

$$\eta_t = \frac{k}{(w + \sum_{i=1}^t \nabla\|f(x_t; \xi_t)\|^2)^{1/3}}, \tag{5}$$

In particular, both the two hyper-parameters $k$ and $w$ (related to $L$ and $G$) are barriers to our parameter-agnostic entry. Consequently, we replace the cubic root term on the denominator in equation 5, and give our choice for the primal variable $x$ and the dual variable $y$ as follows:

$$\alpha_t^x = \sum_{i=1}^t \frac{\|v_i\|^2}{\beta_{i+1}}, \quad \alpha_t^y = \sum_{i=1}^t \frac{\|w_i\|^2}{\beta_{i+1}}. \tag{6}$$

We combine the historical sequence of the estimator and $\beta_t$ to enjoy $\beta_t$'s ability to perceive errors. According to the above choice, we can find that $\eta_t$ is equally capable of handling errors in both subproblems. No matter who generates a larger error, both $\alpha_t^x$ and $\alpha_t^y$ can respond to it and slow down the update. As for separating the updated scale of $x$ and $y$, a consensus idea Lin et al. (2020); Li et al. (2022) requires that the variable $y$ should be updated by a larger learning rate than $x$ to make $y$ to achieve stationary first. Therefore, we should not aggressively update $x$ if the inner maximization sub-problem has not yet been solved accurately. In order to meet the step size automatic adjustment, we use the following strategy to achieve adaptive time scale separation:

$$\eta_t^x = \frac{\gamma}{\max\{\alpha_t^x, \alpha_t^y\}^{1/3}}, \quad \eta_t^y = \frac{\lambda}{(\alpha_t^y)^{1/3}}. \tag{7}$$

Note that $\gamma$ and $\lambda$ aim to ensure the update of $x$ is slower than $y$. In addition, we can properly set their values $\gamma \le \lambda$ to adapt more quickly to different applications. Even if $\gamma = \lambda = 1$, our theorems also hold. Based on the above statement, VRAda can choose the larger historical estimator cumulative value after being adjusted by $\beta_t$. Accordingly, we can make sure that if the inner maximization sub-problem has not yet been solved accurately, the update of $x$ is always slowed down. With the above approach, we can achieve adaptive and scale separation and efficient updating.

## 4 THEORETICAL ANALYSIS

In this section, we present the convergence and sample complexity of our proposed VRAda under Non-Convex-Strongly-Concave (NC-SC) and Non-Convex-Polyak-Łojasiewicz (NC-PL) objectives, respectively. We define $(x, y)$ as an $\epsilon$-stationary point if both $\mathbb{E}\|\nabla_x f(x,y)\| \leq \epsilon$ and $\mathbb{E}\|\nabla_y f(x,y)\| \leq \epsilon$, where the expectation accounts for all algorithmic randomness. As shown in Yang et al. (2022b), this definition of stationarity can be conveniently translated to the near-stationarity of the primal function $\Phi(x) = \max_{y \in \mathcal{Y}} f(x, y)$. Before presenting the theoretical results, we first state some useful assumptions to facilitate our analysis.

**Assumption 1.** *(Smoothness) There exists a constant $L > 0$, such that*

$$\|\nabla f(x_1, y_1) - \nabla f(x_2, y_2)\| \leq L\|(x_1, y_1) - (x_2, y_2)\|,$$

*where $x_1, x_2 \in \mathbb{R}^{d_1}$ and $y_1, y_2 \in \mathcal{Y}$.*

**Assumption 2.** *(Bounded Gradient) For any $x \in \mathbb{R}^{d_1}$ and $y \in \mathcal{Y}$, there exists a constant $G$ such that*

$$\|\nabla_x F(x, y; \xi^x)\| \leq G, \quad \|\nabla_y F(x, y; \xi^y)\| \leq G.$$

**Assumption 3.** *(Bounded Variance) There exists a constant $\sigma$ such that the variance of each gradient estimator is bounded by:*

$$\mathbb{E}[\|\nabla_x f(x, y; \xi^x) - \nabla_x f(x, y)\|^2] \leq \sigma^2, \quad \mathbb{E}[\|\nabla_y f(x, y; \xi^y) - \nabla_y f(x, y)\|^2] \leq \sigma^2,$$

*where $x_1, x_2 \in \mathbb{R}^{d_1}$ and $y_1, y_2 \in \mathcal{Y}$.*

It is worth noting that these assumptions are only presented to facilitate our proof. In the implementation of VRAda, we do not need any information from them to achieve the final result. In equation 1, we represent $y^*(x) := \arg\max_{y \in \mathcal{Y}} f(x, y)$ as the solution of the inner maximization sub-problem, where $y^*(x)$ resides within the interior of $\mathcal{Y}$ for any $x \in \mathbb{R}^{d_1}$. This property ensures that $\nabla_y f(x, y^*(x)) = 0$, which uses the sum of squared norms of past gradients in the denominator. Without this condition, the learning rate would decrease, even in the proximity of the optimal point, resulting in slow convergence. In addition, we aim to find a near stationary point for the minimax problem, denoted by $\mathbb{E}[\|\nabla_x f(x, y)\|] \leq \epsilon$ and $\mathbb{E}[\|\nabla_y f(x, y)\|] \leq \epsilon$, with the expectation encompassing all algorithmic sources of randomness. As such, this stationary notion can be easily translated to the near-stationary of the primal function $\Phi(x) := f(x, y^*(x))$ Yang et al. (2022a;b); Huang et al. (2023); Liu et al. (2023). Accordingly, we introduce an additional assumption as follows:

**Assumption 4.** *(Bounded Primal Function Value) There exists a constant $\Phi_*$ such that for any $x \in \mathbb{R}^{d_1}$, $\Phi(x_t)$ is upper bounded by $\Phi_*$.*

**Remark 1.** Assumptions 1-3 find common application in numerous studies involving adaptive algorithms and minimax optimizations, as evidenced by research such as Carmon et al. (2019); Yang et al. (2020); Levy et al. (2021); Kavis et al. (2022); Huang et al. (2023); Liu et al. (2023). Particularly noteworthy is Assumption 4, which signifies the bounded nature of the domain of $y$-a condition also considered in the analyses of AdaGrad Levy (2017); Levy et al. (2018). In neural networks featuring rectified activations, the scale-invariance property Dinh et al. (2017) renders the imposition of boundedness on $y$ compatible with expressive modeling. Additionally, Wasserstein GANs Arjovsky et al. (2017) utilize critic projections to confine weights within a small cube centered around the origin. Importantly, our proof does not necessitate the assumption of second-order Lipschitz continuity for $y$, setting our proof on more rigorous compared to Li et al. (2022).

### 4.1 ANALYSIS OF THE NC-SC SETTING

We use the following assumption to show the concavity in $y$.

**Assumption 5.** *(Strongly Concave in $y$) Function $f(x, y)$ is $\mu$-strongly-concave ($\mu > 0$) in $y$, that is, for any $x \in \mathbb{R}^{d_1}$ and $y_1, y_2 \in \mathcal{Y}$, we have*

$$f(x, y_1) \geq f(x, y_2) + \langle \nabla_y f(x, y_1), y_1 - y_2 \rangle + \frac{\mu}{2} \|y_1 - y_2\|^2.$$

**Theorem 1.** *Under Assumptions 1-5, VRAda in Algorithm 1 satisfies,*

$$\frac{1}{T}\left[\mathbb{E}\sum_{t=1}^T \|\nabla_x f(x_t, y_t)\| + \mathbb{E}\sum_{t=1}^T \|\nabla_y f(x_t, y_t)\|\right] \leq O(T^{-1/3}).$$

In our proof, we divide the two variables into four cases, i.e., $\mathbb{E}\sum_{t=1}^{T}\|\nabla_x f(x_t, y_t)\|^2$, $\mathbb{E}\sum_{t=1}^{T}\|\epsilon_t^x\|^2$, and $\mathbb{E}\sum_{t=1}^{T}\|\nabla_y f(x_t, y_t)\|^2$, $\mathbb{E}\sum_{t=1}^{T}\|\epsilon_t^y\|^2$, to represent the corresponding accumulative errors and estimators. When the cumulative error term is relatively large, it acts as an upper bound for the cumulative gradient. However, when the accumulated error term is small, we may not establish an upper bound for the cumulative gradient based solely on the error term. In these situations, we can provide additional information to determine the upper bound for the cumulative gradient.

**Remark 2.** The convergence result $O(T^{-1/3})$ in Theorem 1 is summarized among the four cases. If we aim to achieve the $\epsilon$-stationary point by VRAda, the total number of training epochs should satisfy that $T = O(1/\epsilon^{-3})$. In addition, because VRAda only needs two samples, i.e., $O(1)$, to compute estimators and gradients in each training epoch, the total sample complexity is $O(1/\epsilon^{-3})$. It is important to note that both $\gamma$ and $\lambda$ are all constants, i.e., $O(1)$, which does not change the convergence rate $O(T^{-1/3})$. As we mentioned before, simply set the two parameters to 1, and according to the above analysis, Theorem 1 also holds true. The purpose of introducing these two additional parameters is to better adapt to different scenarios. To the best of our knowledge, VRAda outperforms the only existing parameter-agnostic algorithm TiAda Li et al. (2022) in stochastic minimax optimizations, i.e., $O(\epsilon^{-4})$ and aligns the sample complexity to the parametric algorithms Huang et al. (2022; 2023).

**Remark 3.** Note that the sample complexity of TiAda is $O(\epsilon^{-(4+\delta)})$, for any small $\delta > 0$. To achieve the optimal complexity $O(\epsilon^{-4})$, they need to manually set the proper values of separated parameters $\alpha$ and $\beta$, e.g., $\alpha = 0.5 + \delta/(8 + 2\delta)$ and $\beta = 0.5 - \delta/(8 + 2\delta)$. However, if we choose a large $\delta$ value, it cannot achieve the optimal complexity, i.e., $O(\epsilon^{-4})$. In contrast, if we choose a small $\delta$ value, then the adjustment of step size will be very weak. Therefore, it is difficult to balance the adjustment effect of learning rates and the speed of convergence from theoretical and empirical perspectives simultaneously. In Theorem 1, we can see that it does not have any problem-dependent parameters, which indicates that VRAda is fully parameter-agnostic and makes VRAda more stable on real-world machine learning applications.

## 4.2 Analysis of the NC-PL Setting

In the NCPL setting, we investigate the case that the sub-problem in $y$ satisfies the PL condition[2], which is a commonly used condition in Charles & Papailiopoulos (2018); Nouiehed et al. (2019); Xie et al. (2020); Huang et al. (2023). As such, we replace Assumption 5 with the following assumption to indicate the PL condition.

**Assumption 6.** *(PL condition in y) Assume function $f(x, y)$ satisfies $\mu_y$-PL condition in variable $y$ for any fixed $x \in \mathbb{R}^{d_1}$ and $y \in \mathcal{Y}$, such that*

$$\|\nabla_y f(x, y)\|^2 \geq 2\mu_y \left( \max_{y^*} f(x, y^*) - f(x, y) \right). \tag{8}$$

**Theorem 2.** *Under Assumptions 1-4 and 6, VRAda in Algorithm 1 satisfies, after $T$ epochs,*

$$\frac{1}{T}\left[ \mathbb{E}\sum_{t=1}^{T}\|\nabla_x f(x_t, y_t)\| + \mathbb{E}\sum_{t=1}^{T}\|\nabla_y f(x_t, y_t)\| \right] \leq O(T^{-1/3}).$$

In this setting, obtaining a direct upper bound for $\mathbb{E}\sum_{t=1}^{T}\|\nabla_y f(x_t, y_y)\|^2$ proves challenging due to the absence of the strong concavity condition. However, by leveraging the smoothness properties of both variables and the $\mu_y$-PL condition, we can establish an upper bound for $\mathbb{E}\sum_{t=1}^{T}[\Phi(x_t) - f(x_t, y_t)]$. Furthermore, we can transform this into $\mathbb{E}\sum_{t=1}^{T}[\|\nabla_x f(x_t, y_t)\|^2]$ using the quadratic growth condition Karimi et al. (2016), which is the condition is interchangeable with the $\mu_y$-PL condition. It allows us to derive the final result. Therefore, modifying this setting affects solely the upper bound of $\mathbb{E}\sum_{t=1}^{T}\|\nabla_y f(x_t, y_t)\|^2$, which impacts the above case 2 and case 4.

**Remark 4.** Note that VRAda maintains a sample complexity of $O(\epsilon^{-3})$ in Theorem 2, mirroring the result in the NC-SC setting. This demonstrates the scalability of VRAda and underscores the rigor of our proof analysis. In addition, both $\gamma$ and $\lambda$ are constants, and as a result, they do not

---

[2]This commemorates the mathematical Boris Polyak (1935-2023).

change the $O(T^{-1/3})$ convergence rate. To the best of our knowledge, VRAda stands as the inaugural parameter-agnostic algorithm designed specifically for stochastic minimax optimizations to fit the NC-PL setting.

## 5 EXPERIMENTS

In this section, we evaluate the performance of VRAda based on three different scenarios: (1) two test functions with synthetic datasets, (2) optimizing the deep AUC loss (an NC-SC objective) proposed by Yuan et al. (2021), and (3) training the Non-Convex-Non-Concave (NC-NC)[3] Wasserstein-GAN with Gradient Penalty (WGAN-GP) Sinha et al. (2017). Based on the three experiments, we believe that this not only validates our theoretical results but also shows the potential of our proposed VRAda algorithm in real-world scenarios. Additional experimental restuls and setups will be deferred to Appendix A in detail.

### 5.1 TEST FUNCTIONS

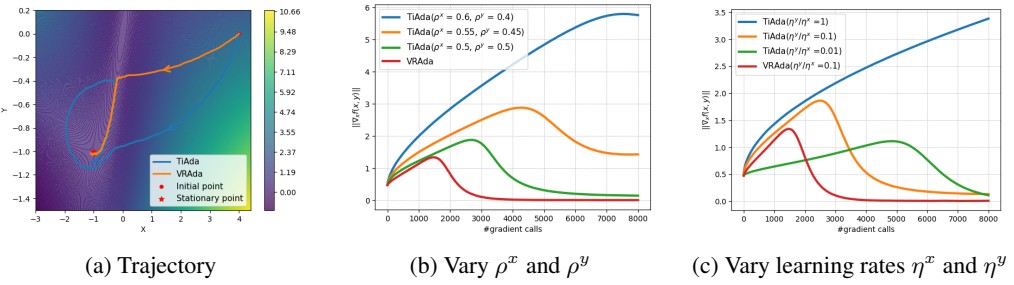

(a) Trajectory      (b) Vary $\rho^x$ and $\rho^y$      (c) Vary learning rates $\eta^x$ and $\eta^y$

Figure 2: Numerical results on the test function $f(x,y) = \log(1 + x^2) - xy + y^2/2$

We test VRAda and TiAda on another function, that is, logarithmic and linear combinations. Similarly, we add a larger noise on the variable $y$ and a smaller noise on the variable $x$. From Figures 2a We can find that VRAda chooses a better route close to the optimal, and that VRAda's radius of convergence is significantly smaller than TiAda. Figure 2b and 2c illustrate TiAda's convergence rate against VRAda on function $f(x,y) = \log(1 + x^2) - xy + y^2/2$, after adjusting the initial learning rate and the parameters $\rho^x$ and $\rho^y$, VRAda consistently outperforms TiAda for both functions.

### 5.2 DEEP AUC

An impactful application of the minimax problem, particularly in the NC-SC setting, is to optimize margin-based min-max surrogate losses. These surrogate losses as practical proxies across various learning scenarios, adeptly capturing the trade-offs inherent in specific objectives. Furthermore, the minimax framework proves valuable in optimizing AUC scores. In situations where imbalanced datasets can skew a model's performance metrics, the optimization of AUC scores has paramount significance. Employing the minimax approach in such contexts not only ensures the model's predictions are accurate but also bolsters their robustness against the challenges posed by data representation disparities. The formulation of the AUC margin Loss Yuan et al. (2021) is as follows:

$$\min_{\mathbf{x} \in \mathbb{R}^{d_1}, (a,b) \in \mathbb{R}^2} \max_{y \in \mathcal{Y}} f(x,a,b,y) := \mathbb{E}_\xi[F(x,a,b,y;\xi)]. \tag{9}$$

We conducted experiments using three distinct image classification datasets: CIFAR10, CIFAR100, and STL10 Elson et al. (2007); Coates et al. (2011); Krizhevsky et al. (2009), all characterized by an imbalance ratio of 10%. The segment of AUC test results is illustrated in Figure 3, clearly showcasing VRAda's consistent superiority over TiAda and TiAda-Adam. For instance, VRAda achieves an impressive AUC score of 0.914 on CIFAR10, while TiAda and TiAda-Adam lag behind at 0.758 and 0.898, respectively. Notably, VRAda's performance advantage over TiAda and TiAda-Adam ranges from approximately 2% to 3% across the various datasets.

---

[3]PL condition is considered as a special case of NC-NC Chen et al. (2022); Huang et al. (2023). Therefore, we use W-GAN to support the efficiency of our VRAda in the NC-PL setting.

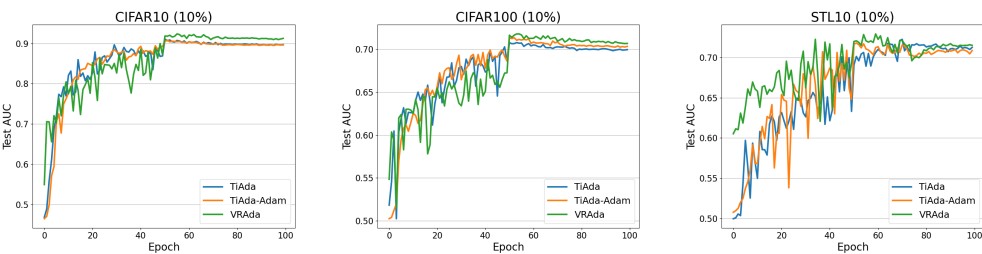

Figure 3: Convergence curves of deep AUC on three datasets with an imbalance ratio of 10%.

## 5.3 WGAN-GP

GANs Arjovsky et al. (2017) highlight the effectiveness of minimax optimization. Here, a discriminator determines if an image is from the dataset while a generator creates samples to fool it. We use the WGAN-GP Sinha et al. (2017) loss in our tests on CIFAR10, CIFAR100, and STL10, ensuring the discriminator functions optimally. Figure 4 shows inception scores on all scenarios.

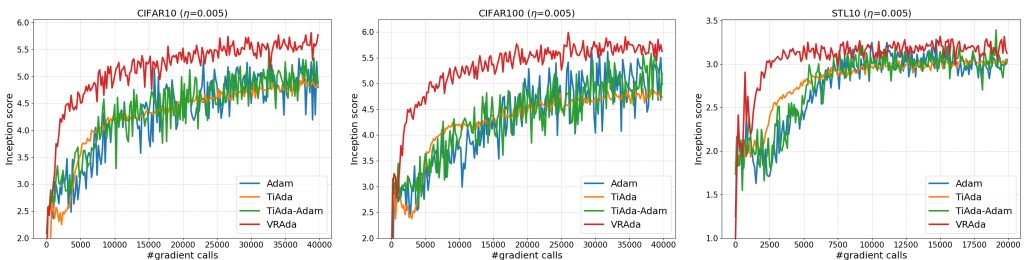

Figure 4: Inception score on WGAN-GP.

During the initial stages of training across all datasets, there is a notable drop in the inception score. This decline is likely attributed to the discriminator rapidly categorizing most generated images as "fake". However, as training progresses, the discriminator becomes increasingly adept at distinguishing real images from generated ones. This feedback loop contributes to the generator's improvement, resulting in an upward trend in the inception score. In our experiments, VRAda achieves an AUC score of 5.65 on CIFAR100, surpassing TiAda, TiAda-Adam, and Adam, which yield scores of 4.69, 4.87, and 4.89, respectively. Notably, VRAda not only outperforms these alternatives but also attains these higher scores more rapidly and consistently as it converges. These results suggest that TiAda may not perform optimally in the NC-NC setting, highlighting VRAda's potential as the first widely applicable parameter-agnostic algorithm for this scenario.

## 6 CONCLUSION

While the parameter-agnostic algorithm TiAda has made significant strides in the realm of minimax optimizations, we have observed its limitations in mitigating the adverse effects of varying inherent variances originating from the primal and dual variables $x$ and $y$. This limitation becomes especially pronounced during the initial stages of training, resulting in degraded learning performance. To address this challenge, our paper introduces VRAda, a variance-reduced adaptive parameter-agnostic algorithm tailored for stochastic minimax optimizations. VRAda effectively reduces the impact of varying inherent variances and autonomously adjusts learning rates after time-scale separation, all without the need for large batches or initial anchor points. Our theoretical analysis demonstrates that VRAda achieves the sample complexity of $O(\epsilon^{-3})$ on both NC-SC and NC-PL settings, surpassing TiAda and aligning with results from existing parametric algorithms. Extensive experimental evidence reinforces the effectiveness and robustness of VRAda across various scenarios, including simple test functions and real-world applications in the NC-SC and NC-NC domains. VRAda holds the potential to bridge the gap between learning theory and empirical practice, offering new opportunities for advancement.

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

# A  ADDITIONAL EXPERIMENTAL

## A.1  RESULTS OF ADDITIONAL TEST FUNCTIONS

In addition to the test functions presented in Sections 1 and 5, we've incorporated two further test results to further validate the robustness and versatility of our VRAda algorithm. We initialized both functions with a starting point of $(-1, -1)$. To emulate the stochastic gradient, we applied Gaussian distribution noise with a mean of $0$ to the function gradient of the primal variable $x$, having a variance of $0.1$, and to the gradient of the dual variable $y$ with a variance of $0.3$. This configuration aligns with the settings used in the aforementioned sections.

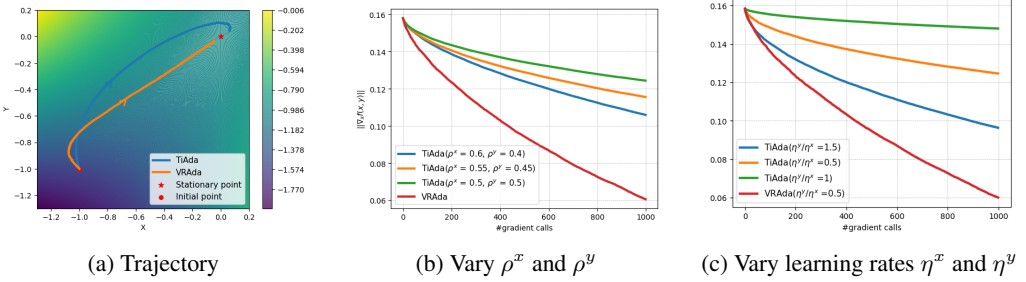

(a) Trajectory      (b) Vary $\rho^x$ and $\rho^y$      (c) Vary learning rates $\eta^x$ and $\eta^y$

Figure 5: Results on the test function $f(x, y) = \cos x - xy$.

From Figure 5, it is evident that VRAda efficiently adjusts the time-scales of $x$ and $y$. Once the appropriate positions are established, they remain constant, leading directly to the optimal solution. In contrast, TiAda continuously adjusts the $x$ and $y$ time-scales throughout the process. The path towards the optimal solution resembles a spiral due to these constant adjustments, resulting in a slower convergence speed.

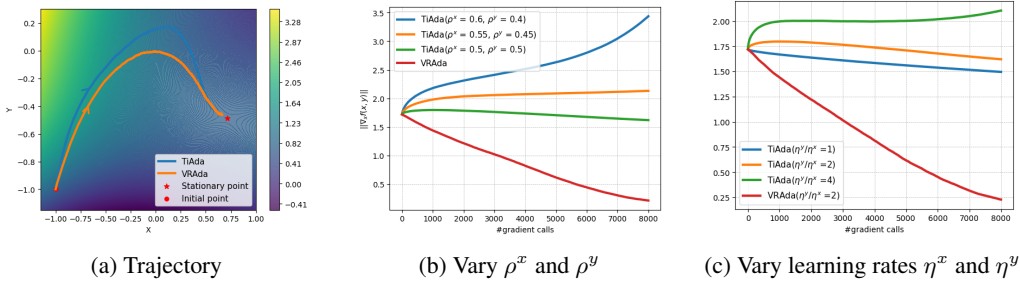

(a) Trajectory      (b) Vary $\rho^x$ and $\rho^y$      (c) Vary learning rates $\eta^x$ and $\eta^y$

Figure 6: Results on the test function $f(x, y) = e^{-x} - xy + y^3$.

Figure 6 reveals that both VRAda and its counterpart are designed to automatically adjust the time scales of $x$ and $y$ to avert divergence. However, VRAda exhibits a superior ability to make precise adjustments, enabling it to navigate towards the optimum solution more efficiently and quickly. This edge in performance is further corroborated by Figure 6b, which underscores the risk of divergence

when manually modulating the force of the separation scale without a comprehensive grasp of the underlying problem.

The robustness of VRAda becomes even more pronounced when additional noise is introduced to the variable $y$. As evidenced in Figure 5a, TiAda, encumbered by noise, demonstrates diminished efficacy in adjusting along the $y$-axis. It overshoots the optimal point before recalibrating its trajectory. A similar pattern is observable in Figure 6a. Despite both algorithms overshooting the optimum on the $y$-axis, VRAda exhibits a commendable resilience. It not only corrects its trajectory more effectively but also converges to the optimal solution with enhanced speed and precision.

## A.2 EXPERIMENTAL SETUPS

### A.2.1 SETUPS OF DEEP AUC

To generate imbalanced data, we utilized the approach described by Yuan et al. (2021). In particular, we divided the training data into two equal portions based on class ID, designating them as positive and negative classes. We then randomly eliminated certain samples from the positive class to create the imbalance, while the testing set remained unchanged. Our experiments were conducted using ResNet20, and we examined imbalance ratios of $5\%$, $10\%$, and $30\%$. For optimization purposes, we executed 100 epochs with a stagewise learning rate. We adjusted the initial stepsize of $x$ within the range [0.1, 0.5] and that of $y$ within [0.6, 1], decaying the rate at the 50% and 75% milestones of the total training epochs across all tests. To streamline the process, we narrowed our parameter search for $\beta_t$ to [0.5, 0.7, 0.9, 0.99]. The batch size was standardized at 128 for all datasets, with the exception of STL10, which was adjusted to 32 due to its smaller size. We implemented a weight decay of $1e$-$4$ consistently across all methodologies. For each dataset, three separate runs were carried out using different random sets (achieved by removing some positive examples with varied random seeds), and we calculated the mean and standard deviations from these results.

### A.2.2 SETUPS OF W-GAN

On CIFAR10 and CIFAR100, we iterated 40,000 times on discriminators and generators, and 2,000 times due to the small dataset of STL10. In this section, we adapted code from Li et al. (2022) for our experiments. For the implementation, we employed a four-layer CNN for the discriminator and another four-layer CNN with transpose convolution layers for the generator, in line with the architecture specified in Daskalakis et al. (2017). We configured the batch size to 512, set the dimension of the latent variable to 50, and assigned a weight of $10^{-4}$ for the gradient penalty term. To further simplify our experiment, we search for our parameter $\beta_t$ in [0.5, 0.7, 0.9, 0.99]. To compute the inception score, we utilized a pre-trained inception network, processing 8000 synthesized samples. For Adam, TiAda and TiAda-Adam, we use the recommended parameters. Since VRAda, TiAda and TiAda-Adam are single-loop algorithms, for fair comparisons, we also update the discriminator only once for each generator update in Adam.

## A.3 ADDITIONAL RESULTS OF DEEP AUC

In Table 1, the outcomes of three imbalance ratios for four optimizers across three datasets are presented. It is evident that VRAda outperforms other optimizers in the majority of scenarios. Specifically, VRAda excels notably in extremely imbalanced settings with an imbalance ratio of $5\%$. In comparisons with TiAda and TiAda-Adam at this imbalance ratio, VRAda demonstrates an improvement of approximately 7% and 5% on the CIFAR10 and CIFAR100 datasets, respectively. It's important to note that as the imbalance ratio rises , the data becomes more balanced, simplifying the classification task. When the imbalance ratio reaches 30%, VRAda's improvement over TiAda and TiAda-Adam stands at about 0.5%, 0.7%, and 0.4% for CIFAR10, CIFAR100, and STL10 datasets, respectively.

## A.4 ADDITIONAL RESULTS OF WGAN-GP

We evaluated three distinct learning rates 0.003, 0.004, and 0.01 for the CIFAR10 and CIFAR100 datasets. However, due to STL10's smaller dataset size, we opted for slightly different learning rates: 0.003, 0.004, and 0.006. Figures 7 through 9 reveal that VRAda consistently outperforms the

Table 1: Testing performance on different datasets

| Datases | imratio | For **AUC** Maximization | | |
| --- | --- | --- | --- | --- |
| | | **5%** | **10%** | **30%** |
| CIFAR10 | Adam | $0.746 \pm 0.018$ | $0.758 \pm 0.014$ | $0.790 \pm 0.013$ |
| | TiAda | $0.828 \pm 0.011$ | $0.898 \pm 0.004$ | $0.937 \pm 0.004$ |
| | TiAda-Adam | $0.853 \pm 0.006$ | $0.896 \pm 0.008$ | $0.935 \pm 0.004$ |
| | **VRAda** | $\mathbf{0.890 \pm 0.003}$ | $\mathbf{0.914 \pm 0.002}$ | $\mathbf{0.940 \pm 0.004}$ |
| CIFAR100 | Adam | $0.605 \pm 0.005$ | $0.614 \pm 0.005$ | $0.619 \pm 0.001$ |
| | TiAda | $0.637 \pm 0.004$ | $0.705 \pm 0.002$ | $0.764 \pm 0.002$ |
| | TiAda-Adam | $0.658 \pm 0.006$ | $0.703 \pm 0.001$ | $0.760 \pm 0.004$ |
| | **VRAda** | $\mathbf{0.672 \pm 0.003}$ | $\mathbf{0.718 \pm 0.006}$ | $\mathbf{0.765 \pm 0.010}$ |
| STL10 | Adam | $0.607 \pm 0.013$ | $0.592 \pm 0.001$ | $0.633 \pm 0.016$ |
| | TiAda | $\mathbf{0.718 \pm 0.002}$ | $0.709 \pm 0.008$ | $0.727 \pm 0.008$ |
| | TiAda-Adam | $0.703 \pm 0.007$ | $0.705 \pm 0.005$ | $0.725 \pm 0.005$ |
| | **VRAda** | $0.706 \pm 0.009$ | $\mathbf{0.726 \pm 0.006}$ | $\mathbf{0.728 \pm 0.007}$ |

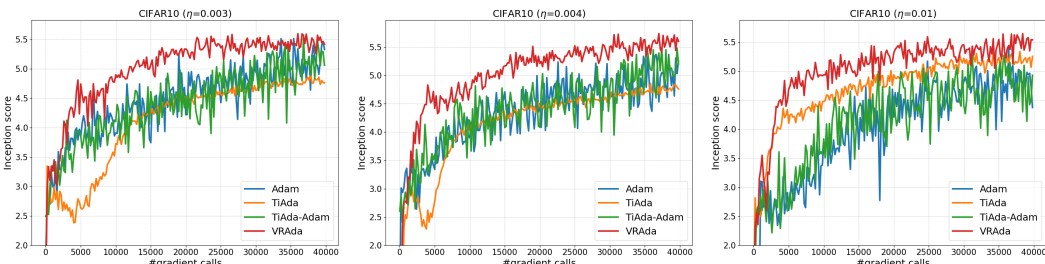

Figure 7: WGAN-GP's Inception score on CIFAR10.

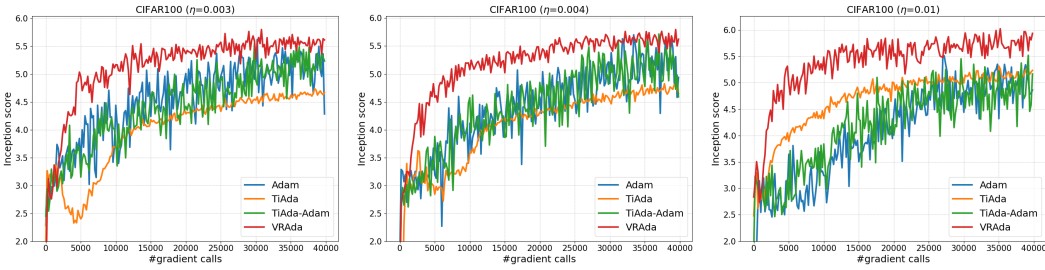

Figure 8: WGAN-GP's Inception score on CIFAR100.

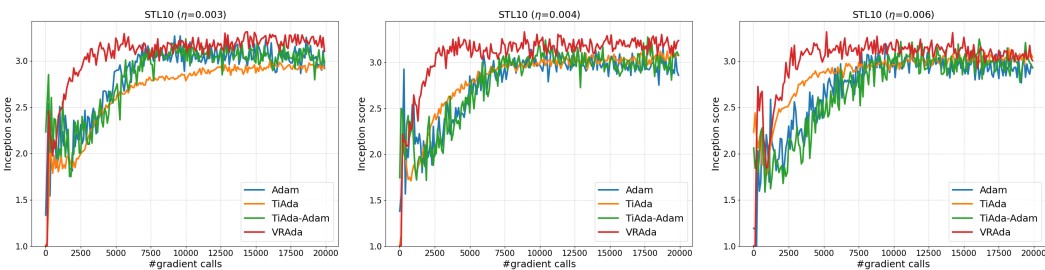

Figure 9: WGAN-GP's Inception score on STL10.

competition across all datasets and parameter configurations. Although a decline in the Inception score can be observed early in training, VRAda rapidly ascends to higher scores, boasting both speed and stability.

For a more precise evaluation of the Inception score, we computed the mean scores of the four optimizers on CIFAR10, specifically averaging over the last 20,000 iterations during the phase when the score stabilized. For CIFAR10 with a learning rate of 0.003, the average scores for Adam, TiAda, TiAda-Adam, and VRAda were 4.89, 4.68, 4.79, and 5.42 respectively. At a learning rate of 0.004, the scores were 4.80, 4.63, 4.79, and 5.45, and at a learning rate of 0.01, they stood at 4.65, 5.10, 4.63, and 5.37. For CIFAR100 with a learning rate of 0.01, the average scores for Adam, TiAda, TiAda-Adam, and VRAda mirrored the previous dataset, being 4.65, 5.10, 4.63, and 5.37, respectively.

## B    USEFUL LEMMAS

**Lemma 1.** *(Lemma A.2 in Yang et al. (2022b)) Let $x_1, \cdots, x_T$ be a sequence of non-negative real numbers, $\alpha \in (0, 1)$, then we have:*

$$\left(\sum_{t=1}^{T} x_t\right)^{1-\alpha} \leq \sum_{t=1}^{T} \frac{x_t}{\left(\sum_{k=1}^{t} x_k\right)^{\alpha}} \leq \frac{1}{1-\alpha} \left(\sum_{t=1}^{T} x_t\right)^{1-\alpha}.$$

**Lemma 2.** *(Lemma A.5 in Nouiehed et al. (2019)) Under Assumptions 1 and 6, we have*

$$\|\nabla\Phi(x_1) - \nabla\Phi(x_2)\| \leq L_\Phi \|x_1 - x_2\|, \ \forall x_1, x_2 \tag{10}$$

*where $L_\Phi = L + \frac{\kappa L}{2}$.*

## C    ANALYSIS OF THEOREM 1

In this section, we reiterate our primary goal of pinpointing a near-stationary point for the minimax problem, represented by $\mathbb{E}[\|\nabla_x f(x, y)\|] \leq \epsilon$ and $\mathbb{E}[\|\nabla_y f(x, y)\|] \leq \epsilon$. Here, the expectation incorporates every element of algorithmic randomness, ensuring a comprehensive and nuanced understanding of the system's behavior amidst varying conditions and inputs.

### C.1    INTERMEDIATE LEMMAS OF THEOREM 1

**Lemma 3.** *Under Assumptions 1-2, the two error dynamics $\mathbb{E}[\sum_{t=1}^{T}\|\epsilon_t^x\|^2]$ and $\mathbb{E}[\sum_{t=1}^{T}\|\epsilon_t^y\|^2]$ can be upper-bounded as follows:*

$$\mathbb{E}\left[\sum_{t=1}^{T}\|\epsilon_t^x\|^2\right]$$

$$\leq \frac{72 + 12G^2}{\beta\tilde{\beta}} + \frac{144L^2\gamma^2}{\beta\tilde{\beta}}\left[\mathbb{E}(\sum_{t=1}^{T}\|v_t\|^2)^{1/3}\right] + \frac{144L^2\lambda^2}{\beta\tilde{\beta}}\left[\mathbb{E}(\sum_{t=1}^{T}\|w_t\|^2)^{1/3}\right] + 6G^2$$

$$+ 144L^2\gamma^2\mathbb{E}\left(1 + \sum_{t=1}^{T}\left(\|\nabla_x f(x_t, y_t; \xi_t^x)\|^2 + \|\nabla_y f(x_t, y_t; \xi_t^y)\|^2\right)\right)^{2/9}\left(\sum_{t=1}^{T}\|v_t\|^2\right)^{1/3}$$

$$+ 144L^2\lambda^2\mathbb{E}\left(1 + \sum_{t=1}^{T}\left(\|\nabla_x f(x_t, y_t; \xi_t^x)\|^2 + \|\nabla_y f(x_t, y_t; \xi_t^y)\|^2\right)\right)^{2/9}\left(\sum_{t=1}^{T}\|w_t\|^2\right)^{1/3}$$

$$+ 36\mathbb{E}\left(1 + \sum_{t=1}^{T}\|\nabla_x f(x_t, y_t; \xi_t^x)\|^2\right)^{1/3},$$

$$\mathbb{E}\sum_{t=1}^{T}\|\epsilon_t^y\|^2$$

$$\leq \frac{72+12G^2}{\beta\tilde{\beta}} + \frac{144L^2\gamma^2}{\beta\tilde{\beta}}\Big[\mathbb{E}(\sum_{t=1}^{T}\|v_t\|^2)^{1/3}\Big] + \frac{144L^2\lambda^2}{\beta\tilde{\beta}}\Big[\mathbb{E}(\sum_{t=1}^{T}\|w_t\|^2)^{1/3}\Big] + 6G^2$$

$$+ 144L^2\gamma^2\Big(1+\sum_{t=1}^{T}\big(\|\nabla_x f(x_t,y_t;\xi_t^x)\|^2 + \|\nabla_y f(x_t,y_t;\xi_t^y)\|^2\big)\Big)^{2/9}\Big(\sum_{t=1}^{T}\|v_t\|^2\Big)^{1/3}$$

$$+ 144L^2\lambda^2\Big(1+\sum_{t=1}^{T}\big(\|\nabla_x f(x_t,y_t;\xi_t^x)\|^2 + \|\nabla_y f(x_t,y_t;\xi_t^y)\|^2\big)\Big)^{2/9}\Big(\sum_{t=1}^{T}\|w_t\|^2\Big)^{1/3}$$

$$+ 36\mathbb{E}\Big(1+\sum_{t=1}^{T}\|\nabla_y f(x_t,y_t;\xi_t^y)\|^2\Big)^{1/3}.$$

*Proof.* Since the error bounds in proving the two are highly similar, we only need to give proof of one of them. Similarly to Levy et al. (2021), we consider the term $\epsilon_t^x$ and split it into the following two terms:

$$\sum_{t=1}^{T}\|\epsilon_t\|^2 = \sum_{t=1}^{\tau^*}\|\epsilon_t\|^2 + \sum_{t=\tau^*+1}^{T}\|\epsilon_t\|^2 \tag{11}$$

Here, $\tau^*$ represents a time-step beyond which we can guarantee $1/\beta_{t+1} - 1/\beta_t \leq 2/3$. The precise definition of $\tau^*$ will be provided subsequently. Next, we present an example to clarify the rationale behind this split.

Taking the square of the above equation and then taking the expectation, we have:

$$\mathbb{E}[\|\epsilon_t^x\|^2]$$
$$\leq (1-\beta_t)^2\mathbb{E}[\|\epsilon_{t-1}^x\|^2] + 2(1-\beta_t)^2\|Z_t^x\|^2 + 2\beta_t^2\mathbb{E}[\|\nabla_x f(x_t,y_t;\xi_t^x) - \nabla_x f(x_t,y_t)\|^2]. \tag{12}$$

Dividing equation 12 by $\beta_t$, and re-arranging implies:

$$\mathbb{E}\sum_{t=1}^{T}\|\epsilon_{t-1}\|^2 \leq -\frac{\mathbb{E}\|\epsilon_T\|^2}{\beta_T} + \sum_{t=1}^{T-1}\Big(\frac{1}{\beta_{t+1}} - \frac{1}{\beta_t}\Big)\mathbb{E}\|\epsilon_t\|^2 + 2\mathbb{E}\sum_{t=1}^{T}\Big[\frac{2(1-\beta_t)^2\|Z_t^x\|^2}{\beta_t}\Big]$$
$$+ 2\sum_{t=1}^{T}\beta_t\|\nabla_x f(x_t,y_t;\xi_t^x) - \nabla_x f(x_t,y_t)\|^2. \tag{13}$$

Drawing from Cutkosky & Orabona (2019), the crucial property of $\beta_t$ is the condition $1/\beta_{t+1} - 1/\beta_t \leq 2/3$ to effectively bound the term $\sum_{t=1}^{T-1}\Big(\frac{1}{\beta_{t+1}} - \frac{1}{\beta_t}\Big)\mathbb{E}\|\epsilon_t\|^2$. However, under Algorithm 1, it's uncertain whether the term $\sum_{t=1}^{T-1}\Big(\frac{1}{\beta_{t+1}} - \frac{1}{\beta_t}\Big)\mathbb{E}\|\epsilon_t\|^2$ fulfills this condition. This uncertainty led us to introduce the split as shown in equation 11.

We now return to the primary proof, aiming to bound the cumulative expectation of errors $\sum_{t=1}^{\tau^*}\|\epsilon_t\|^2$. As derived from equation 12, we have:

$$\sum_{t=1}^{\tau^*}\|\epsilon_t^x\|^2 \leq \sum_{t=1}^{\tau^*}(1-\beta_t)\|\epsilon_{t-1}^x\|^2 + 2\sum_{t=1}^{\tau^*}\|Z_t^x\|^2 + 2\sum_{t=1}^{\tau^*}\beta_t^2\|\nabla_x f(x_t,y_t;\xi_t^x) - \nabla_x f(x_t,y_t)\|^2. \tag{14}$$

Now we define $\beta := \min\{1, 1/G^2\}$, $\tau^* = \max\{t \in [T] : \beta_t \geq \beta\}$. Re-arranging the above and using the definition of $\tau^*$, we have:

$$\beta\sum_{t=1}^{\tau^*}\|\epsilon_t^x\|^2 \leq \|\epsilon_{\tau^*}^x\|^2 + \sum_{t=1}^{\tau^*-1}\beta_{t+1}\|\epsilon_t^x\|^2$$

$$\leq 2\underbrace{\sum_{t=1}^{T}\|Z_t^x\|^2}_{(i)} + 2\underbrace{\sum_{t=1}^{T}\beta_t^2\|\nabla_x f(x_t,y_t;\xi_t^x) - \nabla_x f(x_t,y_t)\|^2}_{(ii)}. \tag{15}$$

Next, we bound the expected value of the above terms.

**Bounding the term** (i). By employing the definition of $Z_t^x$ and invoking Assumption 1, we can deduce that $\|Z_t^x\|^2 \le 8L^2\left((\gamma\eta_{t-1}^x)^2\|v_{t-1}\|^2 + (\lambda\eta_{t-1}^y)^2\|w_{t-1}\|^2\right)$. Further, using the formulation for $\eta_{t-1}^x$ in conjunction with Lemma 1, we can demonstrate the following:

$$
\begin{aligned}
\text{(i)} &\le 8L^2\gamma^2\left(\sum_{t=1}^{T}\frac{\|v_{t-1}\|^2}{\left(\sum_{i=1}^{t}\frac{\|v_i\|^2}{\beta_{i+1}}\right)^{\frac{2}{3}}}\right) + 8L^2\lambda^2\left(\sum_{t=1}^{T}\frac{\|w_{t-1}\|^2}{\left(\sum_{i=1}^{t}\frac{\|w_i\|^2}{\beta_{i+1}}\right)^{\frac{2}{3}}}\right) \\
&\le 8L^2\gamma^2\left(\sum_{t=1}^{T}\frac{\|v_{t-1}\|^2}{\left(\sum_{i=1}^{t}\|v_i\|^2\right)^{\frac{2}{3}}}\right) + 8L^2\lambda^2\left(\sum_{t=1}^{T}\frac{\|w_{t-1}\|^2}{\left(\sum_{i=1}^{t}\|w_i\|^2\right)^{\frac{2}{3}}}\right) \\
&\le 24L^2\gamma^2\left(\sum_{t=1}^{T}\|v_t\|^2\right)^{\frac{1}{3}} + 24L^2\lambda^2\left(\sum_{t=1}^{T}\|w_t\|^2\right)^{\frac{1}{3}},
\end{aligned}
\tag{16}
$$

where the second inequality follows that $\beta_t \le 1$, $\forall t \in [T]$.

**Bounding the term** (ii). Given that $\mathbb{E}[\nabla_x f(x_t, y_t; \xi_t^x)] = \nabla_x f(x_t, y_t)$, we deduce:

$$
\mathbb{E}\left[\beta_t^2\|\nabla_x f(x_t, y_t; \xi_t^x) - \nabla_x f(x_t, y_t)\|^2\right] \le \mathbb{E}\left[\beta_t^2\|\nabla_x f(x_t, y_t; \xi_t^x)\|^2\right].
\tag{17}
$$

Utilizing the above inequality, we can infer:

$$
\begin{aligned}
\mathbb{E}\text{(ii)} &\le \mathbb{E}\sum_{t=1}^{T}\frac{\|\nabla_x f(x_t, y_t; \xi_t^x)\|^2}{(1 + \sum_{i=1}^{t-1}\max\{\|\nabla_x f(x_i, y_i; \xi_i^x)\|^2, \|\nabla_y f(x_i, y_i; \xi_i^y)\|^2\})^{4/3}} \\
&\le \mathbb{E}\sum_{t=1}^{T}\frac{\|\nabla_x f(x_t, y_t; \xi_t^x)\|^2}{(1 + \sum_{i=1}^{t-1}\|\nabla_x f(x_i, y_i; \xi_i^x)\|^2)^{4/3}} \le 12 + 2G^2,
\end{aligned}
\tag{18}
$$

The final inequality is supported by Lemma 6 in Levy et al. (2021), which states that for any sequence of non-negative real numbers $a_1, \cdots, a_n \in [0, a_{\max}]$:

$$
\sum_{i=1}^{n}\frac{a_i}{\left(1 + \sum_{j=1}^{i-1}a_j\right)^{4/3}} \le 12 + 2a_{\max}.
\tag{19}
$$

Combining the above inequalities implies that:

$$
\mathbb{E}\sum_{t=1}^{\tau^*}\|\epsilon_t^x\|^2 \le \frac{24 + 4G^2}{\beta} + \frac{48L^2\gamma^2}{\beta}\left[\mathbb{E}(\sum_{t=1}^{T}\|v_t\|^2)^{1/3}\right] + \frac{48L^2\lambda^2}{\beta}\left[\mathbb{E}(\sum_{t=1}^{T}\|w_t\|^2)^{1/3}\right].
\tag{20}
$$

Next, we bound the term $\sum_{t=1}^{T}\|\epsilon_t\|^2$. Dividing both side of equation 12 by $\sqrt{\beta_t}$, and taking the square, we have:

$$
\frac{\|\epsilon_t^x\|^2}{\beta_t} \le (\frac{1}{\beta_t} - 1)\|\epsilon_{t-1}^x\|^2 + 2\frac{\|Z_t^x\|^2}{\beta_t} + 2\beta_t\|\nabla_x f(x_t, y_t; \xi_t^x) - \nabla_x f(x_t, y_t)\|^2,
\tag{21}
$$

where we use $\beta_t \le 1$ in the above inequality. Re-arranging the above and summing gives,

$$
\begin{aligned}
\sum_{t=1}^{T}\left\|\epsilon_{t-1}^x\right\|^2 &\le \underbrace{-\frac{1}{\beta_T}\|\epsilon_T^x\|^2}_{\text{(iii)}} + \underbrace{\sum_{t=1}^{T}\left(\frac{1}{\beta_{t+1}} - \frac{1}{\beta_t}\right)\|\epsilon_t^x\|^2}_{\text{(iv)}} + \underbrace{2\sum_{t=1}^{T}\frac{\|Z_t^x\|^2}{\beta_t}}_{\text{(v)}} \\
&\quad + \underbrace{2\sum_{t=1}^{T}\beta_t\left\|\nabla_x f(x_t, y_t; \xi_t^x) - \nabla_x f(x_t, y_t)\right\|^2}_{\text{(vi)}}.
\end{aligned}
\tag{22}
$$

Next, we bound the expected value of the above terms.

**Bounding the term** (iii). Since $\beta_T \leq 1$, we have $-\mathbb{E}\|\epsilon_T^x\|^2/\beta_T \leq -\mathbb{E}\|\epsilon_T^x\|^2$.

**Bounding the term** (iv). We first give the upper bound of $\frac{1}{\beta_{t+1}}$. Following Lemma 7 in Levy et al. (2021), given $1/\tilde{\beta} = (1/\beta^{3/2} + G^2)^{2/3}$, we have $1/\beta_{t+1} \leq 1/\tilde{\beta}$, $\forall t \leq \tau^*$ and $1/\beta_{t+1} - 1/\beta_t \leq 2/3$, $\forall t \geq \tau^* + 1$, then we have:

$$
\begin{aligned}
\text{(iv)} &= \sum_{t=1}^{\tau^*} (\frac{1}{\beta_{t+1}} - \frac{1}{\beta_t})\|\epsilon_t^x\|^2 + \sum_{t=\tau^*+1}^{T} (\frac{1}{\beta_{t+1}} - \frac{1}{\beta_t})\|\epsilon_t^x\|^2 \\
&\leq \frac{1}{\tilde{\beta}} \sum_{t=1}^{\tau^*} \|\epsilon_t^x\|^2 + \frac{2}{3} \sum_{t=\tau^*+1}^{T} \|\epsilon_t^x\|^2 \\
&\leq \frac{24 + 4G^2}{\beta\tilde{\beta}} + \frac{48L^2\gamma^2}{\beta\tilde{\beta}}\Big[\mathbb{E}(\sum_{t=1}^{T} \|v_t\|^2)^{1/3}\Big] + \frac{48L^2\lambda^2}{\beta\tilde{\beta}}\Big[\mathbb{E}(\sum_{t=1}^{T} \|w_t\|^2)^{1/3}\Big] + \frac{2}{3} \sum_{t=1}^{T} \|\epsilon_t^x\|^2,
\end{aligned}
$$
(23)

where the last inequality holds by equation 20.

**Bounding the term** (v). Recalling that $\|Z_t^x\|^2 \leq 8L^2\Big((\gamma\eta_{t-1}^x)^2\|v_{t-1}\|^2 + (\lambda\eta_{t-1}^y)^2\|w_{t-1}\|^2\Big)$, using Lemma 1, we have:

$$
\begin{aligned}
&\sum_{t=1}^{T} \frac{\|Z_t^x\|^2}{\beta_t} \\
&\leq 8L^2\gamma^2\Big(\sum_{t=1}^{T} \frac{(\eta_{t-1}^x)^2\|v_{t-1}\|^2}{\beta_t}\Big) + 8L^2\lambda^2\Big(\sum_{t=1}^{T} \frac{(\eta_{t-1}^y)^2\|w_{t-1}\|^2}{\beta_t}\Big) \\
&= 8L^2\gamma^2\Big(\sum_{t=1}^{T} \frac{\|v_{t-1}\|^2/\beta_t}{(\sum_{i=1}^{t-1} \|v_i\|^2/\beta_{i+1})^{2/3}}\Big) + 8L^2\lambda^2\Big(\sum_{t=1}^{T} \frac{\|w_{t-1}\|^2/\beta_t}{(\sum_{i=1}^{t-1} \|w_i\|^2/\beta_{i+1})^{2/3}}\Big) \\
&\leq 24L^2\gamma^2\Big(\sum_{t=1}^{T-1} \|v_t\|^2/\beta_{t+1}\Big)^{1/3} + 24L^2\lambda^2\Big(\sum_{t=1}^{T-1} \|w_t\|^2/\beta_{t+1}\Big)^{1/3} \\
&\leq \frac{24L^2\gamma^2}{(\beta_T)^{1/3}}\Big(\sum_{t=1}^{T-1} \|v_t\|^2\Big)^{1/3} + \frac{24L^2\lambda^2}{(\beta_T)^{1/3}}\Big(\sum_{t=1}^{T-1} \|w_t\|^2\Big)^{1/3} \\
&\leq 24L^2\gamma^2\Big(1 + \sum_{t=1}^{T} \big(\|\nabla_x f(x_t, y_t; \xi_t^x)\|^2 + \|\nabla_y f(x_t, y_t; \xi_t^y)\|^2\big)\Big)^{2/9}\Big(\sum_{t=1}^{T} \|v_t\|^2\Big)^{1/3} \\
&\quad + 24L^2\lambda^2\Big(1 + \sum_{t=1}^{T} \big(\|\nabla_x f(x_t, y_t; \xi_t^x)\|^2 + \|\nabla_y f(x_t, y_t; \xi_t^y)\|^2\big)\Big)^{2/9}\Big(\sum_{t=1}^{T} \|w_t\|^2\Big)^{1/3}.
\end{aligned}
$$
(24)

**Bounding the term** (vi). Note that $\mathbb{E}[\nabla_x f(x_t, y_t; \xi_t^x)] = \nabla_x f(x_t, y_t)$, we have $\mathbb{E}[\beta_t\|\nabla_x f(x_t, y_t; \xi_t^x) - \nabla_x f(x_t, y_t)\|^2] \leq \mathbb{E}[\beta_t\|\nabla_x f(x_t, y_t; \xi_t^x)\|^2]$, then we have:

$$
\begin{aligned}
\mathbb{E}(\text{vi}) &\leq \mathbb{E} \sum_{t=1}^{T} \beta_t\|\nabla_x f(x_t, y_t; \xi_t^x)\|^2 \\
&= \mathbb{E} \sum_{t=1}^{T} \frac{\|\nabla_x f(x_t, y_t; \xi_t^x)\|^2}{(1 + \sum_{i=1}^{t-1} \max\{\|\nabla_x f(x_t, y_t; \xi_t^x)\|^2, \|\nabla_y f(x_t, y_t; \xi_t^y)\|^2\})^{2/3}} \\
&\leq \mathbb{E} \sum_{t=1}^{T} \frac{\|\nabla_x f(x_t, y_t; \xi_t^x)\|^2}{(1 + \sum_{i=1}^{t-1} \|\nabla_x f(x_t, y_t; \xi_t^x)\|^2)^{2/3}} \\
&\leq G^2 + 6\mathbb{E}\Big(1 + \sum_{t=1}^{T} \|\nabla_x f(x_t, y_t; \xi_t^x)\|^2\Big)^{1/3},
\end{aligned}
$$
(25)

where the last inequality holds by Lemma 3 in Levy et al. (2021), i.e., let $b_1, \cdots, b_n \in (0, b]$ be a sequence of non-negative real numbers for some positive real number $b, b_0 > 0$ and $p \in (0, 1]$ a rational number, then,

$$\sum_{i=1}^{n} \frac{b_i}{\left(b_0 + \sum_{j=1}^{i-1} b_j\right)^p} \le \frac{b}{(b_0)^p} + \frac{2}{1-p}\left(b_0 + \sum_{i=1}^{n} b_i\right)^{1-p}. \tag{26}$$

Combining the above inequalities, we have:

$$\frac{1}{3}\mathbb{E}\sum_{t=1}^{T}\|\epsilon_t^x\|^2$$

$$\le \frac{24+4G^2}{\beta\tilde{\beta}} + \frac{48L^2\gamma^2}{\beta\tilde{\beta}}\left[\mathbb{E}(\sum_{t=1}^{T}\|v_t\|^2)^{1/3}\right] + \frac{48L^2\lambda^2}{\beta\tilde{\beta}}\left[\mathbb{E}(\sum_{t=1}^{T}\|w_t\|^2)^{1/3}\right] + 2G^2$$

$$+ 48L^2\gamma^2\mathbb{E}\left(1 + \sum_{t=1}^{T}\left(\|\nabla_x f(x_t, y_t; \xi_t^x)\|^2 + \|\nabla_y f(x_t, y_t; \xi_t^y)\|^2\right)\right)^{2/9}\left(\sum_{t=1}^{T}\|v_t\|^2\right)^{1/3} \tag{27}$$

$$+ 48L^2\lambda^2\mathbb{E}\left(1 + \sum_{t=1}^{T}\left(\|\nabla_x f(x_t, y_t; \xi_t^x)\|^2 + \|\nabla_y f(x_t, y_t; \xi_t^y)\|^2\right)\right)^{2/9}\left(\sum_{t=1}^{T}\|w_t\|^2\right)^{1/3}$$

$$+ 12\mathbb{E}\left(1 + \sum_{t=1}^{T}\|\nabla_x f(x_t, y_t; \xi_t^x)\|^2\right)^{1/3},$$

Similarly, we have:

$$\frac{1}{3}\mathbb{E}\sum_{t=1}^{T}\|\epsilon_t^y\|^2$$

$$\le \frac{24+4G^2}{\beta\tilde{\beta}} + \frac{48L^2\gamma^2}{\beta\tilde{\beta}}\left[\mathbb{E}(\sum_{t=1}^{T}\|v_t\|^2)^{1/3}\right] + \frac{48L^2\lambda^2}{\beta\tilde{\beta}}\left[\mathbb{E}(\sum_{t=1}^{T}\|w_t\|^2)^{1/3}\right] + 2G^2$$

$$+ 48L^2\gamma^2\mathbb{E}\left(1 + \sum_{t=1}^{T}\left(\|\nabla_x f(x_t, y_t; \xi_t^x)\|^2 + \|\nabla_y f(x_t, y_t; \xi_t^y)\|^2\right)\right)^{2/9}\left(\sum_{t=1}^{T}\|v_t\|^2\right)^{1/3} \tag{28}$$

$$+ 48L^2\lambda^2\mathbb{E}\left(1 + \sum_{t=1}^{T}\left(\|\nabla_x f(x_t, y_t; \xi_t^x)\|^2 + \|\nabla_y f(x_t, y_t; \xi_t^y)\|^2\right)\right)^{2/9}\left(\sum_{t=1}^{T}\|w_t\|^2\right)^{1/3}$$

$$+ 12\mathbb{E}\left(1 + \sum_{t=1}^{T}\|\nabla_y f(x_t, y_t; \xi_t^y)\|^2\right)^{1/3}.$$

This completes the proof. $\qquad\square$

**Lemma 4.** *Under Assumptions 1-4 ,term $\sum_{t=1}^{T}\|\nabla_x f(x_t, y_t)\|^2$ can be upper-bounded as follows:*

$$\sum_{t=1}^{T}\|\nabla_x f(x_t, y_t)\|^2 \le \sum_{t=1}^{T}\|\epsilon_t^x\|^2 + \frac{4\Phi_*}{\gamma}(1/\beta_{T+1})^{1/3}(\sum_{t=1}^{T}\|v_t\|^2)^{1/3} + 3L\gamma^2(\sum_{t=1}^{T}\|v_t\|^2)^{1/3}.$$

*Proof.* From Assumption 1 we know that $f(x, y)$ is smooth with respect to $x$, so we have:

$$f(x_{t+1}, y_t) - f(x_t, y_t) \le -\gamma\eta_t^x\langle\nabla_x f(x_t, y_t), v_t\rangle + \frac{L(\gamma\eta_t^x)^2}{2}\|v_t\|^2$$

$$\le -\gamma\eta_t^x\|\nabla_x f(x_t, y_t)\|^2 - \gamma\eta_t^x\langle\nabla_x f(x_t, y_t), \epsilon_t^x\rangle + \frac{L(\gamma\eta_t^x)^2}{2}\|v_t\|^2 \tag{29}$$

$$\le -\frac{\gamma\eta_t^x}{2}\|\nabla_x f(x_t, y_t)\|^2 + \frac{\gamma\eta_t^x}{2}\|\epsilon_t^x\|^2 + \frac{L(\gamma\eta_t^x)^2}{2}\|v_t\|^2.$$

Define $\Delta_1 = f(x_1, y_1)$ and $\forall t \geq 2$,

$$\Delta_t = \begin{cases} f(x_t, y_{t-1}) + f(x_t, y_t), & f(x_t, y_t) \geq f(x_t, y_{t-1}), \\ f(x_t, y_t), & f(x_t, y_t) < f(x_t, y_{t-1}). \end{cases} \tag{30}$$

From Assumption 4 we can get $\Delta_t \leq \Phi(x_t) + \Phi(x_{t-1}) \leq 2\Phi_*$. Re-arranging the above, and summing over $t$, we have:

$$\sum_{t=1}^T \|\nabla_x f(x_t, y_t)\|^2$$

$$\leq \frac{2}{\gamma} \sum_{t=1}^T \frac{1}{\eta_t^x} (f(x_t, y_t) - f(x_{t+1}, y_t)) + \sum_{t=1}^T \|\epsilon_t^x\|^2 + \sum_{t=1}^T L(\gamma\eta_t^x)^2 \|v_t\|^2$$

$$\leq \frac{2}{\gamma} \sum_{t=2}^T (\frac{1}{\eta_t^x} - \frac{1}{\eta_{t-1}^x})\Delta_t - \frac{2\Delta_{T+1}}{\gamma\eta_T^x} + \frac{2\Phi_*}{\gamma\eta_1^x} + \sum_{t=1}^T \|\epsilon_t^x\|^2 + \sum_{t=1}^T L(\gamma\eta_t^x)^2 \|v_t\|^2 \tag{31}$$

$$\leq \sum_{t=1}^T \|\epsilon_t^x\|^2 + \frac{4\Phi_*}{\gamma\eta_T^x} + L\gamma^2 \sum_{t=1}^T \frac{\|v_t\|^2}{(\sum_{i=1}^t \|v_i\|^2)^{2/3}}$$

$$\leq \sum_{t=1}^T \|\epsilon_t^x\|^2 + \frac{4\Phi_*}{\gamma}(1/\beta_{T+1})^{1/3}(\sum_{t=1}^T \|v_t\|^2)^{1/3} + 3L\gamma^2(\sum_{t=1}^T \|v_t\|^2)^{1/3},$$

where the last second inequality holds by $\beta_t < 1$. $\qquad\qquad\square$

Before bounding the term $\mathbb{E} \sum_{t=1}^T \|\nabla_y f(x_t, y_t)\|^2$, we first provide some useful lemmas.

**Lemma 5.** *Given Assumptions 1 to 5, if for $t = t_0$ to $t_1 - 1$ and any $\lambda_t > 0, S_t$,*

$$\|y_{t+1} - y_{t+1}^*\|^2 \leq (1 + \lambda_t)\|y_{t+1} - y_t^*\|^2 + S_t,$$

*then we have:*

$$\mathbb{E}\left[\sum_{t=t_0}^{t_1-1} (f(x_t, y_t^*) - f(x_t, y_t))\right]$$

$$\leq \mathbb{E}\left[\sum_{t=t_0+1}^{t_1-1} \left(\frac{1 - \eta_t^y \mu}{2\eta_t^y}\|y_t - y_t^*\|^2 - \frac{1}{2\eta_t^y(1 + \lambda_t)}\|y_{t+1} - y_{t+1}^*\|^2\right)\right]$$

$$+ \mathbb{E}\left[\sum_{t=t_0}^{t_1-1} \frac{\eta_t^y}{2}\|w_t\|^2\right] + \mathbb{E}\left[\sum_{t=t_0}^{t_1-1} \frac{S_t}{2\eta_t^y(1 + \lambda_t)}\right].$$

*Proof.* For any value of $\lambda_t$, we have:

$$\|y_{t+1} - y_{t+1}^*\|^2 \leq (1 + \lambda_t)\|y_{t+1} - y_t^*\|^2 + S_t$$

$$= (1 + \lambda_t)\|y_t + \eta_t^y w_t - y_t^*\|^2 + S_t$$

$$\leq (1 + \lambda_t)\Big(\|y_t - y_t^*\|^2 + (\eta_t^y)^2\|w_t\|^2 + 2\eta_t^y\langle w_t, y_t - y_t^*\rangle \tag{32}$$

$$+ \eta_t^y \mu\|y_t - y_t^*\|^2 - \eta_t^y \mu\|y_t - y_t^*\|^2\Big) + S_t.$$

Rearranging the terms, we have:

$$\langle w_t, y_t^* - y_t\rangle - \mu\|y_t - y_t^*\|^2$$

$$= \langle \nabla_y f(x_t, y_t; \xi_t^y), y_t^* - y_t\rangle + \langle w_t - \nabla_y f(x_t, y_t; \xi_t^y), y_t^* - y_t\rangle - \frac{\mu}{2}\|y_t - y_t^*\|^2 \tag{33}$$

$$\leq \frac{1 - \mu\eta_t^y}{2\eta_t^y}\|y_t - y_t^*\|^2 - \frac{1}{2\eta_t^y(1 + \lambda_t)}\|y_{t+1} - y_{t+1}^*\|^2 + \frac{\eta_t^y}{2}\|w_t\|^2 + \frac{S_t}{2\eta_t^y(1 + \lambda_t)}.$$

For the term $\langle w_t - \nabla_y f(x_t, y_t; \xi_t^y), y_t^* - y_t \rangle$, we have:

$$
\begin{aligned}
\mathbb{E}\left[w_t - \nabla_y f(x_t, y_t; \xi_t^y)\right] &= (1 - \beta_t)\mathbb{E}\left[w_{t-1} - \nabla_y f(x_{t-1}, y_{t-1}; \xi_t^y)\right] \\
&= (1 - \beta_t)\mathbb{E}\left[w_{t-1} - \nabla_y f(x_{t-1}, y_{t-1})\right] \\
&= (1 - \beta_t)\mathbb{E}\left[w_{t-1} - \nabla_y f(x_{t-1}, y_{t-1}; \xi_{t-1}^y)\right] \\
&\quad\vdots \\
&= (1 - \beta_t)(1 - \beta_{t-1})\cdots(1 - \beta_2)\mathbb{E}\left[w_1 - \nabla_y f(x_1, y_1)\right] \\
&= 0,
\end{aligned}
\tag{34}
$$

then we have:

$$
\mathbb{E}\left[\langle w_t - \nabla_y f(x_t, y_t; \xi_t^y), y_t^* - y_t \rangle\right] = 0.
\tag{35}
$$

Using strongly concave we can get

$$
\mathbb{E}\left[\langle \nabla_y f(x_t, y_t; \xi_t^y), y_t^* - y_t \rangle + \langle w_t - \nabla_y f(x_t, y_t; \xi_t^y), y_t^* - y_t \rangle - \frac{\mu}{2}\|y_t - y_t^*\|^2\right]
$$
$$
\geq (f(x_t, y_t^*) - f(x_t, y_t)).
\tag{36}
$$

Telescoping from $t = t_0$ to $t - 1$, and taking the expectation we finish the proof. $\qquad\square$

**Lemma 6.** *Given Assumptions 1 to 3, we have:*

$$
\mathbb{E}\left[\sum_{t=1}^{T}\left(f\left(x_t, y_t^*\right) - f\left(x_t, y_t\right)\right)\right]
$$
$$
\leq \mathbb{E}\left[\sum_{t=2}^{T}\left(\frac{1 - \eta_t^y \mu}{2\eta_t^y}\|y_t - y_t^*\|^2 - \frac{1}{\eta_t^y(2 + \mu\eta_t^y)}\|y_{t+1} - y_{t+1}^*\|^2\right)\right]
$$
$$
+ \frac{3\lambda}{4}\mathbb{E}\left(\sum_{t=1}^{T}\|w_t\|^2\right)^{2/3} + \frac{3\kappa^2\gamma}{4\lambda}\mathbb{E}\left(\sum_{t=1}^{T}\|v_t\|^2\right)^{2/3} + \frac{\kappa^2\gamma^2}{\lambda^2}\mathbb{E}\left(\sum_{t=1}^{T}\|v_t\|^2\right).
$$

*Proof.* By Young's inequality, we have:

$$
\|y_{t+1} - y_{t+1}^*\|^2 \leq (1 + \lambda_t)\|y_{t+1} - y_t^*\|^2 + \left(1 + \frac{1}{\lambda_t}\right)\|y_{t+1}^* - y_t^*\|^2.
\tag{37}
$$

Then letting $\lambda_t = \frac{\mu\eta_t^y}{2}$ and by Lemma 5, we have:

$$
\mathbb{E}\left[\sum_{t=1}^{T}\left(f\left(x_t, y_t^*\right) - f\left(x_t, y_t\right)\right)\right]
$$
$$
\leq \mathbb{E}\left[\sum_{t=2}^{T}\left(\frac{1 - \eta_t^y \mu}{2\eta_t^y}\|y_t - y_t^*\|^2 - \frac{1}{\eta_t^y(2 + \mu\eta_t^y)}\|y_{t+1} - y_{t+1}^*\|^2\right)\right]
\tag{38}
$$
$$
+ \mathbb{E}\left[\sum_{t=1}^{T}\frac{\eta_t^y}{2}\|w_t\|^2\right] + \mathbb{E}\left[\sum_{t=1}^{T}\frac{(1 + \frac{2}{\mu\eta_t^y})}{\eta_t^y(2 + \mu\eta_t^y)}\|y_{t+1}^* - y_t^*\|^2\right].
$$

We bound the term $\mathbb{E}\left[\sum_{t=1}^{T}\frac{(1 + \frac{2}{\mu\eta_t^y})}{\eta_t^y(2 + \mu\eta_t^y)}\|y_{t+1}^* - y_t^*\|^2\right]$.

$$
\mathbb{E}\left[\sum_{t=1}^{T}\frac{(1 + \frac{2}{\mu\eta_t^y})}{\eta_t^y(2 + \mu\eta_t^y)}\|y_{t+1}^* - y_t^*\|^2\right]
$$
$$
\leq \mathbb{E}\left[\sum_{t=1}^{T}\frac{(1 + \frac{2}{\mu\eta_t^y})}{2\eta_t^y}\|y_{t+1}^* - y_t^*\|^2\right] \leq \kappa^2\mathbb{E}\left[\sum_{t=1}^{T}\frac{(1 + \frac{2}{\mu\eta_t^y})}{2\eta_t^y}(\eta_t^x)^2\|v_t\|^2\right]
\tag{39}
$$
$$
= \kappa^2\mathbb{E}\left[\sum_{t=1}^{T}\left(\frac{(\eta_t^x)^2}{2} + \frac{(\eta_t^x)^2}{\mu(\eta_t^y)^2}\right)\|v_t\|^2\right] = \kappa^2\mathbb{E}\left[\sum_{t=1}^{T}\left(\frac{\gamma}{2\lambda}\eta_t^x + \frac{\gamma^2}{\lambda^2}\right)\|v_t\|^2\right]
$$
$$
\leq \frac{3\kappa^2\gamma}{4\lambda}\mathbb{E}\left(\sum_{t=1}^{T}\|v_t\|^2\right)^{2/3} + \frac{\kappa^2\gamma^2}{\lambda^2}\mathbb{E}\left(\sum_{t=1}^{T}\|v_t\|^2\right).
$$

Combining the above two inequalities, we finish the proof. $\qquad\square$

**Lemma 7.** *Given Assumptions 1 to 3, we have*

$$\mathbb{E}\left[\sum_{t=1}^{T}\left(\frac{1-\eta_t^y\mu}{2\eta_t^y}\left\|y_t-y_t^*\right\|^2-\frac{1}{\eta_t^y\left(2+\mu\eta_t^y\right)}\left\|y_{t+1}-y_{t+1}^*\right\|^2\right)\right]$$
$$\leq\left(\frac{G^{\frac{2}{3}}}{2\lambda}-\frac{\mu}{2}\right)\left\|y_0-y_0^*\right\|^2+\frac{G^2}{\mu^2\eta_T^y}.$$

*Proof.*

$$\mathbb{E}\left[\sum_{t=1}^{T}\left(\frac{1-\eta_t^y\mu}{2\eta_t^y}\left\|y_t-y_t^*\right\|^2-\frac{1}{\eta_t^y\left(2+\mu\eta_t^y\right)}\left\|y_{t+1}-y_{t+1}^*\right\|^2\right)\right]$$
$$\leq\left(\frac{G^{\frac{2}{3}}}{2\lambda}-\frac{\mu}{2}\right)\left\|y_0-y_0^*\right\|^2+\frac{1}{2}\sum_{t=2}^{T-1}\left(\frac{1}{\eta_{t+1}^y}-\frac{1}{\eta_t^y}-\frac{\mu}{2}\right)\left\|y_t-y_t^*\right\|^2$$
$$\leq\left(\frac{G^{\frac{2}{3}}}{2\lambda}-\frac{\mu}{2}\right)\left\|y_0-y_0^*\right\|^2+\frac{1}{2\mu^2}\sum_{t=2}^{T-1}\left(\frac{1}{\eta_{t+1}^y}-\frac{1}{\eta_t^y}-\frac{\mu}{2}\right)\left\|\nabla_y f(x_t,y_t)\right\|^2 \qquad(40)$$
$$\leq\left(\frac{G^{\frac{2}{3}}}{2\lambda}-\frac{\mu}{2}\right)\left\|y_0-y_0^*\right\|^2+\frac{G^2}{2\mu^2}\sum_{t=2}^{T-1}\left(\frac{1}{\eta_{t+1}^y}-\frac{1}{\eta_t^y}-\frac{\mu}{2}\right)$$
$$\leq\left(\frac{G^{\frac{2}{3}}}{2\lambda}-\frac{\mu}{2}\right)\left\|y_0-y_0^*\right\|^2+\frac{G^2}{2\mu^2\eta_T^y}.$$

This completes the proof. $\qquad\square$

**Lemma 8.** *Based on Lemmas 6 and 7, we can upper-bound $\mathbb{E}\left[\sum_{t=1}^{T}\left\|\nabla_y f\left(x_t,y_t\right)\right\|^2\right]$ as follows:*

$$\mathbb{E}\left[\sum_{t=1}^{T}\left\|\nabla_y f\left(x_t,y_t\right)\right\|^2\right]$$
$$\leq\left(\frac{L\kappa G^{\frac{2}{3}}}{\lambda}-\mu L\kappa\right)\left\|y_0-y_0^*\right\|^2+\frac{3\lambda L\kappa}{2}\mathbb{E}\left(\sum_{t=1}^{T}\left\|w_t\right\|^2\right)^{2/3}$$
$$+\frac{9\kappa^3 L\gamma}{2\lambda}\mathbb{E}\left(\sum_{t=1}^{T}\left\|v_t\right\|^2\right)^{2/3}+\frac{3\kappa^3 L\gamma^2}{2\lambda^2}\mathbb{E}\left(\sum_{t=1}^{T}\left\|v_t\right\|^2\right)$$
$$+\frac{L\kappa G^2}{\mu^2}\left(1+\sum_{t=1}^{T}\left(\left\|\nabla_x f(x_t,y_t;\xi_t^x)\right\|^2+\left\|\nabla_y f(x_t,y_t;\xi_t^y)\right\|^2\right)\right)^{2/9}\left(\sum_{t=1}^{T}\left\|w_t\right\|^2\right)^{1/3}.$$

*Proof.* Combining Lemma 6 and 7 we have:

$$\mathbb{E}\left[\sum_{t=1}^{T}\left(f\left(x_t,y_t^*\right)-f\left(x_t,y_t\right)\right)\right]$$
$$\leq\left(\frac{G^{\frac{2}{3}}}{2\lambda}-\frac{\mu}{2}\right)\left\|y_0-y_0^*\right\|^2+\frac{G^2}{2\mu^2\eta_T^y} \qquad(41)$$
$$+\frac{3\lambda}{4}\mathbb{E}\left(\sum_{t=1}^{T}\left\|w_t\right\|^2\right)^{2/3}+\frac{3\kappa^2\gamma}{4\lambda}\mathbb{E}\left(\sum_{t=1}^{T}\left\|v_t\right\|^2\right)^{2/3}+\frac{\kappa^2\gamma^2}{\lambda^2}\mathbb{E}\left(\sum_{t=1}^{T}\left\|v_t\right\|^2\right).$$

According to the $\mu$ strongly concave in Assumption 5, we have:

$$\mathbb{E}\left[\sum_{t=1}^{T}\|\nabla_y f(x_t, y_t)\|^2\right] \leq L^2 \mathbb{E}\left[\sum_{t=1}^{T}\|y_t - y_t^*\|^2\right] \leq 2L\kappa \mathbb{E}\left[\sum_{t=1}^{T}(f(x_t, y_t^*) - f(x_t, y_t))\right] \quad (42)$$

Then we have:

$$\begin{aligned}
&\mathbb{E}\left[\sum_{t=1}^{T}\|\nabla_y f(x_t, y_t)\|^2\right] \\
&\leq \left(\frac{L\kappa G^{\frac{2}{3}}}{\lambda} - \mu L\kappa\right)\|y_0 - y_0^*\|^2 + \frac{3\lambda L\kappa}{2}\mathbb{E}\left(\sum_{t=1}^{T}\|w_t\|^2\right)^{2/3} \\
&\quad + \frac{9\kappa^3 L\gamma}{2\lambda}\mathbb{E}\left(\sum_{t=1}^{T}\|v_t\|^2\right)^{2/3} + \frac{3\kappa^3 L\gamma^2}{2\lambda^2}\mathbb{E}\left(\sum_{t=1}^{T}\|v_t\|^2\right) \\
&\quad + \frac{L\kappa G^2}{\mu^2}\left(1 + \sum_{t=1}^{T}\left(\|\nabla_x f(x_t, y_t; \xi_t^x)\|^2 + \|\nabla_y f(x_t, y_t; \xi_t^y)\|^2\right)\right)^{2/9}\left(\sum_{t=1}^{T}\|w_t\|^2\right)^{1/3}.
\end{aligned} \quad (43)$$

This completes the proof. $\qquad\qquad\square$

## C.2 PROOF OF THEOREM 1

Now, we come to the proof of Theorem 1.

*Proof.* For simplicity, we denote $\theta = \max\{\gamma, \lambda\}$. Due to equation 34, we have $\mathbb{E}[w_t - \nabla_y f(x_t, y_t; \xi_t^y)] = 0$, similarly, we have $\mathbb{E}[v_t - \nabla_x f(x_t, y_t; \xi_t^x)] = 0$, then we have $\|v_t\|^2 \leq \|\nabla_x f(x_t, y_t)\|^2 + \|\epsilon_t^x\|^2$ and $\|w_t\|^2 \leq \|\nabla_y f(x_t, y_t)\|^2 + \|\epsilon_t^y\|^2$, we divide the final part of the proof into four subcases:

**Case 1:** Assume $\mathbb{E}\sum_{t=1}^{T}\|\nabla_x f(x_t, y_t)\|^2 \leq 6\mathbb{E}\sum_{t=1}^{T}\|\epsilon_t^x\|^2$ and $\mathbb{E}\sum_{t=1}^{T}\|\nabla_y f(x_t, y_t)\|^2 \leq 6\mathbb{E}\sum_{t=1}^{T}\|\epsilon_t^y\|^2$. Using the condition of this subcase implies

$$\mathbb{E}\sum_{t=1}^{T}(\|v_t\|^2 + \|w_t\|^2) \leq 7\mathbb{E}\sum_{t=1}^{T}(\|\epsilon_t^x\|^2 + \|\epsilon_t^y\|^2). \quad (44)$$

Combining equation 27 and equation 28, we have:

$$\begin{aligned}
&\frac{1}{3}\mathbb{E}\sum_{t=1}^{T}(\|\epsilon_t^x\|^2 + \|\epsilon_t^y\|^2) \\
&\leq \frac{48 + 8G^2}{\beta\tilde{\beta}} + \frac{96L^2\gamma^2}{\beta\tilde{\beta}}\left[\mathbb{E}(\sum_{t=1}^{T}\|v_t\|^2)^{1/3}\right] + \frac{96L^2\lambda^2}{\beta\tilde{\beta}}\left[\mathbb{E}(\sum_{t=1}^{T}\|w_t\|^2)^{1/3}\right] + 4G^2 \\
&\quad + \underbrace{96L^2\gamma^2\left(1 + \sum_{t=1}^{T}\left(\|\nabla_x f(x_t, y_t; \xi_t^x)\|^2 + \|\nabla_y f(x_t, y_t; \xi_t^y)\|^2\right)\right)^{2/9}\left(\sum_{t=1}^{T}\|v_t\|^2\right)^{1/3}}_{(I)} \\
&\quad + \underbrace{96L^2\lambda^2\left(1 + \sum_{t=1}^{T}\left(\|\nabla_x f(x_t, y_t; \xi_t^x)\|^2 + \|\nabla_y f(x_t, y_t; \xi_t^y)\|^2\right)\right)^{2/9}\left(\sum_{t=1}^{T}\|w_t\|^2\right)^{1/3}}_{(II)} \\
&\quad + 24\mathbb{E}\left(1 + \sum_{t=1}^{T}\left(\|\nabla_x f(x_t, y_t; \xi_t^x)\|^2 + \|\nabla_y f(x_t, y_t; \xi_t^y)\|^2\right)\right)^{1/3}
\end{aligned} \quad (45)$$

According to Young's inequality, for any $a, b > 0$, and $p, q > 1 : \frac{1}{p} + \frac{1}{q} = 1$ we have $ab \leq a^p/p + b^q/q$. Setting $p = 3/2$, $q = 3$, we have

$$a^{2/9} b^{1/3} = \left(a\rho^{9/2}\right)^{2/9} \left(b/\rho^3\right)^{1/3} \leq \frac{\left(a\rho^{9/2}\right)^{2p/9}}{p} + \frac{\left(b/\rho^3\right)^{q/3}}{q} = \frac{2}{3} a^{1/3}\rho^{3/2} + \frac{b}{3\rho^3}. \quad (46)$$

Setting $\rho = (1344L^2\gamma^2)^{1/3}$ for Term (I) and $\rho = (1344L^2\lambda^2)^{1/3}$ for Term (II) we have:

$$
\begin{aligned}
&\frac{1}{3}\mathbb{E}\sum_{t=1}^{T}(\|\epsilon_t^x\|^2 + \|\epsilon_t^y\|^2) \\
&\leq \frac{48 + 8G^2}{\beta\tilde{\beta}} + \frac{96L^2\gamma^2}{\beta\tilde{\beta}}\left[\mathbb{E}(\sum_{t=1}^{T}\|v_t\|^2)^{1/3}\right] + \frac{96L^2\lambda^2}{\beta\tilde{\beta}}\left[\mathbb{E}(\sum_{t=1}^{T}\|w_t\|^2)^{1/3}\right] + 4G^2 \\
&\quad + (24 + 2347L^3(\gamma^3 + \lambda^3))\left(1 + \mathbb{E}\sum_{t=1}^{T}\left(\|\nabla_x f(x_t, y_t; \xi_t^x)\|^2 + \|\nabla_y f(x_t, y_t; \xi_t^y)\|^2\right)\right)^{1/3} \\
&\quad + \frac{1}{42}\mathbb{E}\sum_{t=1}^{T}\|v_t\|^2 + \frac{1}{42}\mathbb{E}\sum_{t=1}^{T}\|w_t\|^2.
\end{aligned}
\quad (47)
$$

Re-arranging and using Case 1, we have:

$$
\begin{aligned}
&\frac{1}{6}\mathbb{E}\sum_{t=1}^{T}(\|\epsilon_t^x\|^2 + \|\epsilon_t^y\|^2) \\
&\leq \frac{48 + 8G^2}{\beta\tilde{\beta}} + \frac{96L^2\gamma^2}{\beta\tilde{\beta}}\left[\mathbb{E}(\sum_{t=1}^{T}\|v_t\|^2)^{1/3}\right] + \frac{96L^2\lambda^2}{\beta\tilde{\beta}}\left[\mathbb{E}(\sum_{t=1}^{T}\|w_t\|^2)^{1/3}\right] + 4G^2 \\
&\quad + (24 + 2347L^3(\gamma^3 + \lambda^3))\left(1 + \mathbb{E}\sum_{t=1}^{T}\left(\|\nabla_x f(x_t, y_t; \xi_t^x)\|^2 + \|\nabla_y f(x_t, y_t; \xi_t^y)\|^2\right)\right)^{1/3} \\
&\leq \frac{48 + 8G^2}{\beta\tilde{\beta}} + \frac{192L^2\theta^2}{\beta\tilde{\beta}}\left(7\mathbb{E}\sum_{t=1}^{T}(\|\epsilon_t^x\|^2 + \|\epsilon_t^y\|^2)\right)^{1/3} + 4G^2 \\
&\quad + (24 + 2347L^3(\gamma^3 + \lambda^3))\left(1 + 2\sigma^2 T + \sum_{t=1}^{T}\left(\|\nabla_x f(x_t, y_t)\|^2 + \|\nabla_y f(x_t, y_t)\|^2\right)\right)^{1/3} \\
&\leq \frac{48 + 8G^2}{\beta\tilde{\beta}} + \frac{192L^2\theta^2}{\beta\tilde{\beta}}\left(7\mathbb{E}\sum_{t=1}^{T}(\|\epsilon_t^x\|^2 + \|\epsilon_t^y\|^2)\right)^{1/3} + 4G^2 \\
&\quad + (24 + 2347L^3(\gamma^3 + \lambda^3))\left(1 + 2\sigma^2 T + 6\sum_{t=1}^{T}\left(\|\epsilon_t^x\|^2 + \|\epsilon_t^y\|^2\right)\right)^{1/3}
\end{aligned}
\quad (48)
$$

Above implies,

$$\mathbb{E}\sum_{t=1}^{T}\|\nabla_x f(x_t, y_t)\|^2 + \mathbb{E}\sum_{t=1}^{T}\|\nabla_y f(x_t, y_t)\|^2 \leq 6\mathbb{E}\sum_{t=1}^{T}(\|\epsilon_t^x\|^2 + \|\epsilon_t^y\|^2) \leq O\left(T^{1/3}\right). \quad (49)$$

**Case 2:** Assume $\mathbb{E}\sum_{t=1}^{T}\|\nabla_x f(x_t, y_t)\|^2 \leq 6\mathbb{E}\sum_{t=1}^{T}\|\epsilon_t^x\|^2$ and $\mathbb{E}\sum_{t=1}^{T}\|\nabla_y f(x_t, y_t)\|^2 \geq 6\mathbb{E}\sum_{t=1}^{T}\|\epsilon_t^y\|^2$. Combining equation 27 and equation 43 we have:

$$
\frac{1}{3}\mathbb{E}\sum_{t=1}^{T}\|\epsilon_t^x\|^2 + \frac{\lambda^2}{54\kappa^3 L\gamma^2}\mathbb{E}\sum_{t=1}^{T}\|\nabla_y f(x_t, y_t)\|^2
$$

$$
\leq \underbrace{\frac{24 + 4G^2}{\beta\tilde{\beta}} + 2G^2 + \frac{\lambda^2\left(\frac{L\kappa G^{\frac{2}{3}}}{\lambda} - \mu L\kappa\right)}{27\kappa^2 L^2\gamma^2}\Phi_*}_{C_1}
$$

$$
+ \frac{96L^2\theta^2}{\beta\tilde{\beta}}\left(\mathbb{E}\sum_{t=1}^{T}\left(\|v_t\|^2 + \|w_t\|^2\right)\right)^{1/3} + \underbrace{\left(\frac{3\lambda^3 L}{108\kappa^2\gamma^2} + \frac{\lambda}{108\gamma}\right)}_{C_2}\left(\mathbb{E}\sum_{t=1}^{T}\left(\|v_t\|^2 + \|w_t\|^2\right)\right)^{2/3}
$$

$$
+ \underbrace{48L^2\gamma^2\mathbb{E}\left(1 + \sum_{t=1}^{T}\left(\|\nabla_x f(x_t, y_t; \xi_t^x)\|^2 + \|\nabla_y f(x_t, y_t; \xi_t^y)\|^2\right)\right)^{2/9}\left(\sum_{t=1}^{T}\|v_t\|^2\right)^{1/3}}_{\text{(III)}}
$$

$$
+ \underbrace{\left(48L^2\lambda^2 + \frac{G^2\lambda^2}{54L^2\gamma^2}\right)\left(1 + \sum_{t=1}^{T}\left(\|\nabla_x f(x_t, y_t; \xi_t^x)\|^2 + \|\nabla_y f(x_t, y_t; \xi_t^y)\|^2\right)\right)^{2/9}\left(\sum_{t=1}^{T}\|w_t\|^2\right)^{1/3}}_{\text{(IV)}}
$$

$$
+ 12\mathbb{E}\left(1 + \sum_{t=1}^{T}\|\nabla_x f(x_t, y_t; \xi_t^x)\|^2\right)^{1/3} + \frac{1}{36}\mathbb{E}\left(\sum_{t=1}^{T}\|v_t\|^2\right).
$$

(50)

According to equation 46, letting $C_3 = 48L^2\lambda^2 + \frac{G^2\lambda^2}{54L^2\gamma^2}$, setting $\rho = (2016L^2\gamma^2)^{1/3}$ for Term (III) and $\rho = (42C_3\kappa^3 L\gamma^2/\lambda^2)^{1/3}$ for Term (IV) we have:

$$
\frac{1}{3}\mathbb{E}\sum_{t=1}^{T}\|\epsilon_t^x\|^2 + \frac{\lambda^2}{54\kappa^3 L\gamma^2}\mathbb{E}\sum_{t=1}^{T}\|\nabla_y f(x_t, y_t)\|^2
$$

$$
\leq C_1 + \frac{96L^2\theta^2}{\beta\tilde{\beta}}\left(\mathbb{E}\sum_{t=1}^{T}\left(\|v_t\|^2 + \|w_t\|^2\right)\right)^{1/3} + C_2\left(\mathbb{E}\sum_{t=1}^{T}\left(\|v_t\|^2 + \|w_t\|^2\right)\right)^{2/3}
$$

$$
+ \underbrace{\left(1437L^3\gamma^3 + \frac{5C_3^{\frac{2}{3}}\kappa\gamma L^{\frac{1}{2}}}{\lambda} + 12\right)}_{C_4}\mathbb{E}\left(1 + \sum_{t=1}^{T}\left(\|\nabla_x f(x_t, y_t; \xi_t^x)\|^2 + \|\nabla_y f(x_t, y_t; \xi_t^y)\|^2\right)\right)^{1/3}
$$

$$
+ \frac{\lambda^2}{126\kappa^3 L\gamma^2}\mathbb{E}\sum_{t=1}^{T}\|w_t\|^2 + \frac{1}{36}\mathbb{E}\left(\sum_{t=1}^{T}\|v_t\|^2\right) + \frac{1}{126}\mathbb{E}\sum_{t=1}^{T}\|v_t\|^2.
$$

(51)

Using Case 2 implies

$$
\mathbb{E}\sum_{t=1}^{T}\|w_t\|^2 \leq \mathbb{E}\sum_{t=1}^{T}\|\nabla_y f(x_t, y_t)\|^2 + \mathbb{E}\sum_{t=1}^{T}\|\epsilon_t^y\|^2 \leq \frac{7}{6}\mathbb{E}\sum_{t=1}^{T}\|\nabla_y f(x_t, y_t)\|^2. \tag{52}
$$

Then, we have:

$$
\begin{aligned}
\frac{1}{12}\mathbb{E}\sum_{t=1}^{T} & \|\epsilon_t^x\|^2 + \frac{\lambda^2}{108\kappa^3 L\gamma^2}\mathbb{E}\sum_{t=1}^{T}\|\nabla_y f(x_t, y_t)\|^2 \\
& \leq C_1 + \frac{192 L^2\theta^2}{\beta\tilde{\beta}}\left(7\mathbb{E}\sum_{t=1}^{T}\left(\|\epsilon_t^x\|^2 + \|\nabla_y f(x_t, y_t)\|^2\right)\right)^{1/3} \\
& \quad + 2C_2\left(7\mathbb{E}\sum_{t=1}^{T}\left(\|\epsilon_t^x\|^2 + \|\nabla_y f(x_t, y_t)\|^2\right)\right)^{2/3} \\
& \quad + C_4\mathbb{E}\left(1 + 2\sigma^2 T + 6\mathbb{E}\sum_{t=1}^{T}\left(\|\epsilon_t^x\|^2 + \|\nabla_y f(x_t, y_t)\|^2\right)\right)^{1/3}
\end{aligned}
\tag{53}
$$

Denote $m_1 = \min\{\frac{1}{12}, \frac{\lambda^2}{108\kappa^3 L\gamma^2}\}$, we have:

$$
\begin{aligned}
2\mathbb{E}\sum_{t=1}^{T}\|\epsilon_t^x\|^2 & + 2\mathbb{E}\sum_{t=1}^{T}\|\nabla_y f(x_t, y_t)\|^2 \\
& \leq \frac{2C_1}{m_1} + \frac{384 L^2\theta^2}{m_1\beta\tilde{\beta}}\left(7\mathbb{E}\sum_{t=1}^{T}(\|\epsilon_t^x\|^2 + \|\nabla_y f(x_t, y_t)\|^2)\right)^{1/3} \\
& \quad + \frac{4C_2}{m_1}\left(7\mathbb{E}\sum_{t=1}^{T}\left(\|\epsilon_t^x\|^2 + \|\nabla_y f(x_t, y_t)\|^2\right)\right)^{2/3} \\
& \quad + \frac{2C_4}{m_1}\left(1 + 2\sigma^2 T + 6\mathbb{E}\sum_{t=1}^{T}(\|\epsilon_t^x\|^2 + \|\nabla_y f(x_t, y_t)\|^2)\right)^{1/3}.
\end{aligned}
\tag{54}
$$

It implies that:

$$
\mathbb{E}\sum_{t=1}^{T}\|\nabla_x f(x_t, y_t)\|^2 + \mathbb{E}\sum_{t=1}^{T}\|\nabla_y f(x_t, y_t)\|^2 \leq 6\mathbb{E}\sum_{t=1}^{T}\left(\|\epsilon_t^x\|^2 + \|\nabla_y f(x_t, y_t)\|^2\right) \leq O(T^{1/3}).
\tag{55}
$$

**Case 3:** Assume $\mathbb{E}\sum_{t=1}^{T}\|\nabla_x f(x_t, y_t)\|^2 \geq 6\mathbb{E}\sum_{t=1}^{T}\|\epsilon_t^x\|^2$ and $\mathbb{E}\sum_{t=1}^{T}\|\nabla_y f(x_t, y_t)\|^2 \leq 6\mathbb{E}\sum_{t=1}^{T}\|\epsilon_t^y\|^2$. Following equation 31 we have:

$$
\begin{aligned}
\sum_{t=1}^{T} & \|\nabla_x f(x_t, y_t)\|^2 \\
& \leq \sum_{t=1}^{T}\|\epsilon_t^x\|^2 + \frac{4\Phi_*}{\gamma}(1/\beta_{T+1})^{1/3}(\sum_{t=1}^{T}\|v_t\|^2)^{1/3} + 3L\gamma^2(\sum_{t=1}^{T}\|v_t\|^2)^{1/3} \\
& \leq \sum_{t=1}^{T}\|\epsilon_t^x\|^2 + 3L\gamma^2(\sum_{t=1}^{T}\|v_t\|^2)^{1/3} \\
& \quad + \frac{4\Phi_*}{\gamma}\left(1 + \sum_{t=1}^{T}\left(\|\nabla_x f(x_t, y_t; \xi_t^x)\|^2 + \|\nabla_y f(x_t, y_t; \xi_t^y)\|^2\right)\right)^{2/9}\left(\sum_{t=1}^{T}\|v_t\|^2\right)^{1/3}.
\end{aligned}
\tag{56}
$$

Combining equation 28 and equation 56 we have:

$$\frac{1}{3}\mathbb{E}\sum_{t=1}^{T}\|\epsilon_t^y\|^2 + \mathbb{E}\sum_{t=1}^{T}\|\nabla_x f(x_t,y_t)\|^2$$

$$\leq \frac{24+4G^2}{\beta\tilde{\beta}} + \frac{48L^2\gamma^2}{\beta\tilde{\beta}}\Big[\mathbb{E}(\sum_{t=1}^{T}\|v_t\|^2)^{1/3}\Big] + \frac{48L^2\lambda^2}{\beta\tilde{\beta}}\Big[\mathbb{E}(\sum_{t=1}^{T}\|w_t\|^2)^{1/3}\Big] + 2G^2$$

$$+ \underbrace{(48L^2\gamma^2 + \frac{4\Phi_*}{\gamma})\Big(1 + \sum_{t=1}^{T}\big(\|\nabla_x f(x_t,y_t;\xi_t^x)\|^2 + \|\nabla_y f(x_t,y_t;\xi_t^y)\|^2\big)\Big)^{2/9}\Big(\sum_{t=1}^{T}\|v_t\|^2\Big)^{1/3}}_{(V)}$$

$$+ \underbrace{48L^2\lambda^2\Big(1 + \sum_{t=1}^{T}\big(\|\nabla_x f(x_t,y_t;\xi_t^x)\|^2 + \|\nabla_y f(x_t,y_t;\xi_t^y)\|^2\big)\Big)^{2/9}\Big(\sum_{t=1}^{T}\|w_t\|^2\Big)^{1/3}}_{(VI)}$$

$$+ 12\mathbb{E}\Big(1 + \sum_{t=1}^{T}\|\nabla_y f(x_t,y_t;\xi_t^y)\|^2\Big)^{1/3} + \sum_{t=1}^{T}\|\epsilon_t^x\|^2 + 3L\gamma^2(\sum_{t=1}^{T}\|v_t\|^2)^{1/3}$$

(57)

According to equation 46, letting $C_5 = 48L^2\gamma^2 + \frac{4\Phi_*}{\gamma}$, setting $\rho = (\frac{2C_5}{3})^{1/3}$ for Term (V) and $\rho = (1344L^2\lambda^2)^{1/3}$ for Term (VI) we have:

$$\frac{1}{3}\mathbb{E}\sum_{t=1}^{T}\|\epsilon_t^y\|^2 + \mathbb{E}\sum_{t=1}^{T}\|\nabla_x f(x_t,y_t)\|^2$$

$$\leq (12 + C_5^{\frac{3}{2}} + 1174L^3\lambda^3)\Big(1 + \sum_{t=1}^{T}\big(\|\nabla_x f(x_t,y_t;\xi_t^x)\|^2 + \|\nabla_y f(x_t,y_t;\xi_t^y)\|^2\big)\Big)^{1/3}$$

$$+ \frac{24+4G^2}{\beta\tilde{\beta}} + \Big(\frac{96L^2\theta^2}{\beta\tilde{\beta}} + 3L\gamma^2\Big)\Big(\mathbb{E}\sum_{t=1}^{T}(\|v_t\|^2 + \|w_t\|^2)\Big)^{1/3} + \sum_{t=1}^{T}\|\epsilon_t^x\|^2$$

$$+ \frac{1}{2}\Big(\sum_{t=1}^{T}\|v_t\|^2\Big) + \frac{1}{84}\Big(\sum_{t=1}^{T}\|w_t\|^2\Big).$$

(58)

Using Case 3 implies

$$\mathbb{E}\sum_{t=1}^{T}\|v_t\|^2 \leq \mathbb{E}\sum_{t=1}^{T}\|\nabla_x f(x_t,y_t)\|^2 + \mathbb{E}\sum_{t=1}^{T}\|\epsilon_t^x\|^2 \leq \frac{7}{6}\mathbb{E}\sum_{t=1}^{T}\|\nabla_x f(x_t,y_t)\|^2. \tag{59}$$

Then we have

$$\mathbb{E}\sum_{t=1}^{T}\|\epsilon_t^y\|^2 + \mathbb{E}\sum_{t=1}^{T}\|\nabla_x f(x_t,y_t)\|^2$$

$$\leq (12 + C_5^{\frac{3}{2}} + 1174L^3\lambda^3)\Big(1 + 2\sigma^2 T + 6\mathbb{E}\sum_{t=1}^{T}\big(\|\epsilon_t^y\|^2 + \|\nabla_x f(x_t,y_t)\|^2\big)\Big)^{1/3} \tag{60}$$

$$+ \frac{24+4G^2}{\beta\tilde{\beta}} + \Big(\frac{96L^2\theta^2}{\beta\tilde{\beta}} + 3L\gamma^2\Big)\Big(\mathbb{E}\sum_{t=1}^{T}6\big(\|\epsilon_t^y\|^2 + \|\nabla_x f(x_t,y_t)\|^2\big)\Big)^{1/3}.$$

It implies that:

$$\mathbb{E}\sum_{t=1}^{T}\|\nabla_x f(x_t,y_t)\|^2 + \mathbb{E}\sum_{t=1}^{T}\|\nabla_y f(x_t,y_t)\|^2 \leq 6\mathbb{E}\sum_{t=1}^{T}\big(\|\epsilon_t^y\|^2 + \|\nabla_x f(x_t,y_t)\|^2\big) \leq O(T^{1/3}).$$

(61)

**Case 4:** Assume $\mathbb{E} \sum_{t=1}^{T} \|\nabla_x f(x_t, y_t)\|^2 \geq 6\mathbb{E} \sum_{t=1}^{T} \|\epsilon_t^x\|^2$ and $\mathbb{E} \sum_{t=1}^{T} \|\nabla_y f(x_t, y_t)\|^2 \geq 6\mathbb{E} \sum_{t=1}^{T} \|\epsilon_t^y\|^2$. Following equation 43 and equation 56 we have:

$$
\begin{aligned}
\sum_{t=1}^{T} & \|\nabla_x f(x_t, y_t)\|^2 + \sum_{t=1}^{T} \|\nabla_y f(x_t, y_t)\|^2 \\
\leq & \left( \frac{L\kappa}{\eta_1^y} - \mu L \kappa \right) \|y_0 - y_0^*\|^2 + \frac{3\lambda L \kappa}{2} \mathbb{E} \left( \sum_{t=1}^{T} \|w_t\|^2 \right)^{2/3} + \sum_{t=1}^{T} \|\epsilon_t^x\|^2 \\
& + \frac{9\kappa^3 L \gamma}{2\lambda} \mathbb{E} \left( \sum_{t=1}^{T} \|v_t\|^2 \right)^{2/3} + \frac{3\kappa^3 L \gamma^2}{2\lambda^2} \mathbb{E} \left( \sum_{t=1}^{T} \|v_t\|^2 \right) + 3L\gamma^2 (\sum_{t=1}^{T} \|v_t\|^2)^{1/3} \\
& + \frac{L\kappa G^2}{\mu^2} \left( 1 + \sum_{t=1}^{T} \left( \|\nabla_x f(x_t, y_t; \xi_t^x)\|^2 + \|\nabla_y f(x_t, y_t; \xi_t^y)\|^2 \right) \right)^{2/9} \left( \sum_{t=1}^{T} \|w_t\|^2 \right)^{1/3} \\
& + \frac{4\Phi_*}{\gamma} \left( 1 + \sum_{t=1}^{T} \left( \|\nabla_x f(x_t, y_t; \xi_t^x)\|^2 + \|\nabla_y f(x_t, y_t; \xi_t^y)\|^2 \right) \right)^{2/9} \left( \sum_{t=1}^{T} \|v_t\|^2 \right)^{1/3} \\
\leq & \left( \frac{2\kappa^2 G^{\frac{2}{3}}}{\lambda} - 2L\kappa \right) \Phi_* + 3L\gamma^2 \left( \mathbb{E} \sum_{t=1}^{T} (\|v_t\|^2 + \|w_t\|^2) \right)^{1/3} \\
& + \left( \frac{3\lambda L \kappa}{2} + \frac{9\kappa^3 L \gamma}{2\lambda} \right) \left( \mathbb{E} \sum_{t=1}^{T} (\|v_t\|^2 + \|w_t\|^2) \right)^{2/3} + \sum_{t=1}^{T} \|\epsilon_t^x\|^2 \\
& + \underbrace{\frac{G^2}{\mu^2} \left( 1 + \sum_{t=1}^{T} \left( \|\nabla_x f(x_t, y_t; \xi_t^x)\|^2 + \|\nabla_y f(x_t, y_t; \xi_t^y)\|^2 \right) \right)^{2/9} \left( \sum_{t=1}^{T} \|w_t\|^2 \right)^{1/3}}_{\text{(VII)}} \\
& + \underbrace{\frac{4\Phi_*}{\gamma} \left( 1 + \sum_{t=1}^{T} \left( \|\nabla_x f(x_t, y_t; \xi_t^x)\|^2 + \|\nabla_y f(x_t, y_t; \xi_t^y)\|^2 \right) \right)^{2/9} \left( \sum_{t=1}^{T} \|v_t\|^2 \right)^{1/3}}_{\text{(VIII)}}
\end{aligned}
\tag{62}
$$

According to equation 46, setting $\rho = (7G^2/6\mu^2)^{1/3}$ for Term (VII) and $\rho = (29\Phi_*/9\gamma)^{1/3}$ for Term (VIII) we have:

$$
\begin{aligned}
\sum_{t=1}^{T} & \|\nabla_x f(x_t, y_t)\|^2 + \sum_{t=1}^{T} \|\nabla_y f(x_t, y_t)\|^2 \\
\leq & \left( \frac{2\kappa^2 G^{\frac{2}{3}}}{\lambda} - 2L\kappa \right) \Phi_* + 3L\gamma^2 \left( \mathbb{E} \sum_{t=1}^{T} (\|v_t\|^2 + \|w_t\|^2) \right)^{1/3} + \frac{2}{7} \sum_{t=1}^{T} \|v_t\|^2 \\
& + \left( \frac{3\lambda L \kappa}{2} + \frac{9\kappa^3 L \gamma}{2\lambda} \right) \left( \mathbb{E} \sum_{t=1}^{T} (\|v_t\|^2 + \|w_t\|^2) \right)^{2/3} + \sum_{t=1}^{T} \|\epsilon_t^x\|^2 + \frac{3}{7} \sum_{t=1}^{T} \|w_t\|^2 \\
& + \left( \frac{4G^3}{3\mu^3} + \frac{16\Phi_*^{\frac{3}{2}}}{\gamma^{\frac{3}{2}}} \right) \left( 1 + \sum_{t=1}^{T} \left( \|\nabla_x f(x_t, y_t; \xi_t^x)\|^2 + \|\nabla_y f(x_t, y_t; \xi_t^y)\|^2 \right) \right)^{1/3}.
\end{aligned}
\tag{63}
$$

Using Case 4 implies that:

$$
\sum_{t=1}^{T} \|\nabla_x f(x_t, y_t)\|^2 + \sum_{t=1}^{T} \|\nabla_y f(x_t, y_t)\|^2
$$

$$
\leq \left( \frac{4\kappa^2 G^{\frac{2}{3}}}{\lambda} - 4L\kappa \right) \Phi_* + 6L\gamma^2 \left( 2\mathbb{E} \sum_{t=1}^{T} (\|\nabla_x f(x_t, y_t)\|^2 + \|\nabla_y f(x_t, y_t)\|^2) \right)^{1/3}
$$

$$
+ \left( 3\lambda L\kappa + \frac{9\kappa^3 L\gamma}{\lambda} \right) \left( 2\mathbb{E} \sum_{t=1}^{T} (\|\nabla_x f(x_t, y_t)\|^2 + \|\nabla_y f(x_t, y_t)\|^2) \right)^{2/3}
$$

$$
+ \left( \frac{8G^3}{3\mu^3} + \frac{32\Phi_*^{\frac{3}{2}}}{\gamma^{\frac{3}{2}}} \right) \left( 1 + 2\sigma^2 T + \sum_{t=1}^{T} \left( \|\nabla_x f(x_t, y_t)\|^2 + \|\nabla_y f(x_t, y_t)\|^2 \right) \right)^{1/3}
$$

$$(64)$$

It implies that:

$$
\left[ \mathbb{E} \sum_{t=1}^{T} \|\nabla_x f(x_t, y_t)\|^2 + \mathbb{E} \sum_{t=1}^{T} \|\nabla_y f(x_t, y_t)\|^2 \right] \leq O(T^{1/3}).
\tag{65}
$$

Conclude all the above cases, using Cauchy-Schwarz inequality, we can easily get

$$
\frac{1}{T} \left[ \mathbb{E} \sum_{t=1}^{T} \|\nabla_x f(x_t, y_t)\| + \mathbb{E} \sum_{t=1}^{T} \|\nabla_y f(x_t, y_t)\| \right]
$$

$$
\leq \frac{\sqrt{2}}{\sqrt{T}} \left[ \sqrt{\mathbb{E} \sum_{t=1}^{T} \|\nabla_x f(x_t, y_t)\|^2 + \mathbb{E} \sum_{t=1}^{T} \|\nabla_y f(x_t, y_t)\|^2} \right] \leq O(\frac{1}{T^{1/3}})
\tag{66}
$$

This completes the proof. $\qquad\square$

## D  ANALYSIS OF THEOREM 2

In this section, we will replace Assumption 5 with Assumption 6. We present a revised upper bound for $\mathbb{E} \sum_{t=1}^{T} \|\nabla_y f(x_t, y_t)\|^2$, taking into account the $\mu_y$-PL condition.

### D.1  INTERMEDIATE LEMMA OF THEOREM 2

**Lemma 9.** *Under Assumption 1, 2 and 6, we have*

$$
\mathbb{E} \sum_{t=1}^{T} \|\nabla_y f(x_t, y_t)\|^2 \leq (16\kappa^2 L^2 + 2\kappa L L_\Phi + \frac{2\kappa L\lambda}{G^{\frac{2}{3}}}) \mathbb{E} \sum_{t=1}^{T} (\eta_t^x)^2 \|v_t\|^2.
\tag{67}
$$

*Proof.* Using the smoothness of $f(x, \cdot)$ we have:

$$
f(x_{t+1}, y_t) \leq f(x_{t+1}, y_{t+1}) - \eta_t^y \langle \nabla_y f(x_{t+1}, y_t), w_t \rangle + \frac{L}{2} \|y_{t+1} - y_t\|^2.
\tag{68}
$$

For the term $-\eta_t^y \langle \nabla_y f(x_{t+1}, y_t), w_t \rangle$, we have

$$
- \eta_t^y \langle \nabla_y f(x_{t+1}, y_t), w_t \rangle
$$

$$
\leq -\frac{\eta_t^y}{2} \left( \|\nabla_y f(x_{t+1}, y_t)\|^2 + \|w_t\|^2 - \|\nabla_y f(x_{t+1}, y_t) - \nabla_y f(x_t, y_t) + \nabla_y f(x_t, y_t) - w_t\|^2 \right)
$$

$$
\leq -\frac{\eta_t^y}{2} \|\nabla_y f(x_{t+1}, y_t)\|^2 - \frac{\eta_t^y}{2} \|w_t\|^2 + \frac{\eta_t^y L^2}{2} \|x_{t+1} - x_t\|^2 + \frac{\eta_t^y}{2} \|\nabla_y f(x_t, y_t) - w_t\|^2
$$

$$
\leq -\eta_t^y \mu_y \left( \Phi(x_{t+1}) - f(x_{t+1}, y_t) \right) - \frac{\eta_t^y}{2} \|w_t\|^2 + \frac{\eta_t^y L^2}{2} \|x_{t+1} - x_t\|^2 + \frac{\eta_t^y}{2} \|\nabla_y f(x_t, y_t) - w_t\|^2,
\tag{69}
$$

where the last inequality holds by $\mu_y$-PL condition. Then we have

$$
\begin{aligned}
f(x_{t+1}, y_t) \leq{} & f(x_{t+1}, y_{t+1}) - \eta_t^y \mu_y \left( \Phi(x_{t+1}) - f(x_{t+1}, y_t) \right) - \frac{\eta_t^y}{2} \|w_t\|^2 \\
& + \frac{\eta_t^y L^2}{2} \|x_{t+1} - x_t\|^2 + \frac{\eta_t^y}{2} \|\nabla_y f(x_t, y_t) - w_t\|^2.
\end{aligned}
\tag{70}
$$

Rearranging the above, we have:

$$
\begin{aligned}
\Phi(x_{t+1}) - f(x_{t+1}, y_{t+1}) \leq{} & (1 - \mu_y \eta_t^y) \left( \Phi(x_{t+1}) - f(x_{t+1}, y_t) \right) - \frac{\eta_t^y}{2} \|w_t\|^2 \\
& + \frac{\eta_t^y L^2}{2} \|x_{t+1} - x_t\|^2 + \frac{\eta_t^y}{2} \|\nabla_y f(x_t, y_t) - w_t\|^2.
\end{aligned}
\tag{71}
$$

Next, using smoothness of $f(\cdot, y)$, we have:

$$
f(x_t, y_t) + \langle \nabla_x f(x_t, y_t), x_{t+1} - x_t \rangle - \frac{L}{2} \|x_{t+1} - x_t\|^2 \leq f(x_{t+1}, y_t).
\tag{72}
$$

Then we have

$$
\begin{aligned}
& f(x_t, y_t) - f(x_{t+1}, y_t) \\
\leq{} & -\langle \nabla_x f(x_t, y_t), x_{t+1} - x_t \rangle + \frac{L}{2} \|x_{t+1} - x_t\|^2 \\
={} & \eta_t^x \langle \nabla_x f(x_t, y_t) - \nabla \Phi(x_t), v_t \rangle - \langle \nabla \Phi(x_t), x_{t+1} - x_t \rangle + \frac{L}{2} \|x_{t+1} - x_t\|^2 \\
\leq{} & \eta_t^x \omega_t \|\nabla \Phi(x_t) - \nabla_x f(x_t, y_t)\|^2 + \frac{\eta_t^x}{\omega_t} \|v_t\|^2 + \Phi(x_t) - \Phi(x_{t+1}) + \frac{(\eta_t^x)^2 L_\Phi}{2} \|v_t\|^2 + \frac{L(\eta_t^x)^2}{2} \|v_t\|^2 \\
\leq{} & L^2 \omega_t \eta_t^x \|y_t - y_t^*\|^2 + \frac{\eta_t^x}{\omega_t} \|v_t\|^2 + \Phi(x_t) - \Phi(x_{t+1}) + L_\Phi (\eta_t^x)^2 \|v_t\|^2 \\
\leq{} & \frac{2 L^2 \omega_t \eta_t^x}{\mu} \left( \Phi(x_t) - f(x_t, y_t) \right) + \frac{\eta_t^x}{\omega_t} \|v_t\|^2 + \Phi(x_t) - \Phi(x_{t+1}) + L_\Phi (\eta_t^x)^2 \|v_t\|^2,
\end{aligned}
\tag{73}
$$

where the two inequality hold by $L < L_\Phi$ and the last two inequality holds by smoothness of $\Phi(x_t)$, and the parameter $\omega_t$ will be determined later. Then we have

$$
\begin{aligned}
\Phi(x_{t+1}) - f(x_{t+1}, y_{t+1}) ={} & \Phi(x_{t+1}) - \Phi(x_t) + \Phi(x_t) - f(x_t, y_t) + f(x_t, y_t) - f(x_{t+1}, y_t) \\
\leq{} & (1 - \mu_y \eta_t^y)(1 + \frac{2 L^2 \omega_t \eta_t^x}{\mu}) \left( \Phi(x_t) - f(x_t, y_t) \right) \\
& + (1 - \mu_y \eta_t^y)(\frac{\eta_t^x}{\omega_t} + L_\Phi (\eta_t^x)^2) \|v_t\|^2 - \frac{\eta_t^y}{2} \|w_t\|^2 \\
& + \frac{\eta_t^y L^2}{2} \|x_{t+1} - x_t\|^2 + \frac{\eta_t^y}{2} \|\nabla_y f(x_t, y_t) - w_t\|^2.
\end{aligned}
\tag{74}
$$

Because $\mathbb{E}[w_t] = \nabla_y f(x_t, y_t)$, so we have $\mathbb{E}[\|w_t - \nabla_y f(x_t, y_t)\|^2] \leq \mathbb{E}[\|w_t\|^2]$. If $\eta_t^y \geq \frac{1}{\mu}$ for $t = 1, \cdots, t = t_0$, then we have

$$
\mathbb{E} \sum_{t=2}^{t_0+1} \left[ (\Phi(x_t) - f(x_t, y_t)) \right] \leq \mathbb{E} \sum_{t=1}^{t_0} \frac{\eta_t^y L^2}{2} \|x_{t+1} - x_t\|^2,
\tag{75}
$$

Now we consider $t = t_0, \cdots, T$. Rearranging the above and summing up, we also have:

$$
\begin{aligned}
& \mathbb{E} \sum_{t=t_0+1}^{T} \left( \mu \eta_t^y + 2 L^2 \omega_t \eta_t^x (\eta_t^y - \frac{1}{\mu}) \right) \left( \Phi(x_t) - f(x_t, y_t) \right) \\
& \leq \mathbb{E} \sum_{t=t_0}^{T} (1 - \mu_y \eta_t^y)(\frac{\eta_t^x}{\omega_t} + L_\Phi (\eta_t^x)^2) \|v_t\|^2 + \mathbb{E} \sum_{t=t_0}^{T} \frac{\eta_t^y L^2}{2} \|x_{t+1} - x_t\|^2.
\end{aligned}
\tag{76}
$$

Setting $\omega_t = \frac{1}{4L^2\eta_t^x(\frac{1}{\mu}-\eta_t^y)}$, we have $\mu\eta_t^y + 2L^2\omega_t\eta_t^x(\eta_t^y - \frac{1}{\mu}) \geq \frac{1}{2}$, and $(1-\mu_y\eta_t^y)(1+\frac{2L^2\omega_t\eta_t^x}{\mu}) \leq (4\kappa L + L_\Phi)(\eta_t^x)^2$ for $t > t_0$. Then we have

$$\frac{1}{2}\mathbb{E}\sum_{t=t_0+1}^T [(\Phi(x_t) - f(x_t,y_t))]$$

$$\leq (4\kappa L + L_\Phi)\mathbb{E}\sum_{t=t_0}^T(\eta_t^x)^2\|v_t\|^2 + \mathbb{E}\sum_{t=t_0}^T \frac{\eta_t^y L^2}{2}\|x_{t+1}-x_t\|^2. \tag{77}$$

Summing above two cases, we have

$$\mathbb{E}\sum_{t=1}^T[\Phi(x_t)-f(x_t,y_t)] \leq (8\kappa L + 2L_\Phi + \eta_1^y)\mathbb{E}\sum_{t=1}^T(\eta_t^x)^2\|v_t\|^2$$

$$\leq (8\kappa L + 2L_\Phi + \frac{\lambda}{G^{\frac{2}{3}}})\mathbb{E}\sum_{t=1}^T(\eta_t^x)^2\|v_t\|^2 \tag{78}$$

From Karimi et al. (2016), we know a function is L-smooth and satisfies PL conditions with constant $\mu_y$, it also satisfies the quadratic growth (QG) condition. Using QG growth we have:

$$\|\nabla_y(x_t,y_t)\|^2 \leq L^2\|y_t^* - y_t\|^2 \leq 2\kappa L(\Phi(x_t) - f(x_t,y_t)). \tag{79}$$

Then we have

$$\mathbb{E}\sum_{t=1}^T \|\nabla_y f(x_t,y_t)\|^2 \leq (16\kappa^2 L^2 + 2\kappa LL_\Phi + \frac{2\kappa L\lambda}{G^{\frac{2}{3}}})\mathbb{E}\sum_{t=1}^T(\eta_t^x)^2\|v_t\|^2. \tag{80}$$

$\square$

## D.2 PROOF OF THEOREM 2

If we change the Assumption from strongly concave to $\mu$-PL condition, this will only affect Case 2 and Case 4. In the following part, we give the new version of the bound.

Case 2: Assume $\mathbb{E}\sum_{t=1}^T \|\nabla_x f(x_t,y_t)\|^2 \leq 6\mathbb{E}\sum_{t=1}^T\|\epsilon_t^x\|^2$ and $\mathbb{E}\sum_{t=1}^T\|\nabla_y f(x_t,y_t)\|^2 \geq 6\mathbb{E}\sum_{t=1}^T\|\epsilon_t^y\|^2$. Combining equation 27 and equation 80, we have:

$$\frac{1}{3}\mathbb{E}\sum_{t=1}^T\|\epsilon_t^x\|^2 + \mathbb{E}\sum_{t=1}^T\|\nabla_y f(x_t,y_t)\|^2$$

$$\leq \frac{24+4G^2}{\beta\tilde{\beta}} + \frac{48L^2\gamma^2}{\beta\tilde{\beta}}\Big[\mathbb{E}(\sum_{t=1}^T\|v_t\|^2)^{1/3}\Big] + \frac{48L^2\lambda^2}{\beta\tilde{\beta}}\Big[\mathbb{E}(\sum_{t=1}^T\|w_t\|^2)^{1/3}\Big] + 2G^2$$

$$+ \underbrace{48L^2\gamma^2\mathbb{E}\Big(1+\sum_{t=1}^T\big(\|\nabla_x f(x_t,y_t;\xi_t^x)\|^2 + \|\nabla_y f(x_t,y_t;\xi_t^y)\|^2\big)\Big)^{2/9}\Big(\sum_{t=1}^T\|v_t\|^2\Big)^{1/3}}_{(P_1)}$$

$$+ \underbrace{48L^2\lambda^2\mathbb{E}\Big(1+\sum_{t=1}^T\big(\|\nabla_x f(x_t,y_t;\xi_t^x)\|^2 + \|\nabla_y f(x_t,y_t;\xi_t^y)\|^2\big)\Big)^{2/9}\Big(\sum_{t=1}^T\|w_t\|^2\Big)^{1/3}}_{(P_2)}$$

$$+ 12\mathbb{E}\Big(1+\sum_{t=1}^T\|\nabla_x f(x_t,y_t;\xi_t^x)\|^2\Big)^{1/3} + (16\kappa^2 L^2 + 2\kappa LL_\Phi + \frac{2\kappa L\lambda}{G^{\frac{2}{3}}})\mathbb{E}\sum_{t=1}^T(\eta_t^x)^2\|v_t\|^2. \tag{81}$$

According to equation 46, setting $\rho = (1344L^2\gamma^2)^{1/3}$ for Term $(P_1)$ and $\rho = (150L^2\lambda^2)^{1/3}$ for Term $(P_2)$ we have:

$$
\frac{1}{3}\mathbb{E}\sum_{t=1}^{T}\|\epsilon_t^x\|^2 + \mathbb{E}\sum_{t=1}^{T}\|\nabla_y f(x_t, y_t)\|^2
$$

$$
\leq \frac{24 + 4G^2}{\beta\tilde{\beta}} + 2G^2 + \frac{1}{72}\mathbb{E}\sum_{t=1}^{T}\|v_t\|^2 + \frac{3}{28}\mathbb{E}\sum_{t=1}^{T}\|w_t\|^2
$$

$$
+ (\frac{96L^2\theta^2}{\beta\tilde{\beta}} + 16\kappa^2 L^2 + 2\kappa L L_\Phi + \frac{2\kappa L\lambda}{G^{\frac{2}{3}}})\left(\mathbb{E}\sum_{t=1}^{T}\left(\|v_t\|^2 + \|w_t\|^2\right)\right)^{1/3}
$$

$$
+ (12 + 1174L^3\gamma^3 + 392L^3\lambda^3)\left(1 + \sum_{t=1}^{T}(\|\nabla_x f(x_t, y_t; \xi_t^x)\|^2 + \|\nabla_y f(x_t, y_t; \xi_t^y)\|^2)\right)^{1/3}
$$

(82)

Using Case 2, we have:

$$
\frac{1}{4}\mathbb{E}\sum_{t=1}^{T}\|\epsilon_t^x\|^2 + \frac{1}{4}\mathbb{E}\sum_{t=1}^{T}\|\nabla_y f(x_t, y_t)\|^2
$$

$$
\leq \frac{24 + 4G^2}{\beta\tilde{\beta}} + 2G^2 + 4\kappa L + \frac{2\kappa\Phi_*}{L}
$$

$$
+ (\frac{96L^2\theta^2}{\beta\tilde{\beta}} + 16\kappa^2 L^2 + 2\kappa L L_\Phi + \frac{2\kappa L\lambda}{G^{\frac{2}{3}}})\left(7\mathbb{E}\sum_{t=1}^{T}\left(\|\epsilon_t^x\|^2 + \|\nabla_y(x_t, y_t)\|^2\right)\right)^{1/3}
$$

$$
+ (12 + 1174L^3\gamma^3 + 392L^3\lambda^3)\left(1 + 2\sigma^2 T + \mathbb{E}\sum_{t=1}^{T}(\|\epsilon_t^x\|^2 + \|\nabla_y f(x_t, y_t)\|^2)\right)^{1/3}
$$

(83)

It implies that

$$
\mathbb{E}\sum_{t=1}^{T}\|\nabla_x f(x_t, y_t)\|^2 + \mathbb{E}\sum_{t=1}^{T}\|\nabla_y f(x_t, y_t)\|^2 \leq 6\mathbb{E}\sum_{t=1}^{T}\|\epsilon_t^x\|^2 + 6\mathbb{E}\sum_{t=1}^{T}\|\nabla_y f(x_t, y_t)\|^2 \leq O(T^{1/3}).
$$

(84)

Case 4: Assume $\mathbb{E}\sum_{t=1}^{T}\|\nabla_x f(x_t, y_t)\|^2 \geq 6\mathbb{E}\sum_{t=1}^{T}\|\epsilon_t^x\|^2$ and $\mathbb{E}\sum_{t=1}^{T}\|\nabla_y f(x_t, y_t)\|^2 \geq 6\mathbb{E}\sum_{t=1}^{T}\|\epsilon_t^y\|^2$. Following equation 56 and equation 80 we have:

$$
\sum_{t=1}^{T}\|\nabla_x f(x_t, y_t)\|^2 + \sum_{t=1}^{T}\|\nabla_y f(x_t, y_t)\|^2
$$

$$
\leq \sum_{t=1}^{T}\|\epsilon_t^x\|^2 + (16\kappa^2 L^2 + 2\kappa L L_\Phi + \frac{2\kappa L\lambda}{G^{\frac{2}{3}}} + 3L\gamma^2)\mathbb{E}\sum_{t=1}^{T}(\eta_t^x)^2\|v_t\|^2
$$

$$
+ \underbrace{\frac{4\Phi_*}{\gamma}\left(1 + \sum_{t=1}^{T}\left(\|\nabla_x f(x_t, y_t; \xi_t^x)\|^2 + \|\nabla_y f(x_t, y_t; \xi_t^y)\|^2\right)\right)^{2/9}\left(\sum_{t=1}^{T}\|v_t\|^2\right)^{1/3}}_{(P_3)}.
$$

(85)

According to equation 46, setting $\rho = (\frac{14\Phi_*}{3\gamma})^{1/3}$ for Term $(P_3)$ we have

$$\sum_{t=1}^{T} \|\nabla_x f(x_t, y_t)\|^2 + \sum_{t=1}^{T} \|\nabla_y f(x_t, y_t)\|^2$$

$$\leq \sum_{t=1}^{T} \|\epsilon_t^x\|^2 + \underbrace{(16\kappa^2 L^2 + 2\kappa L L_\Phi + \frac{2\kappa L\lambda}{G^{\frac{2}{3}}} + 3L\gamma^2)}_{C_6} \mathbb{E} \sum_{t=1}^{T} (\eta_t^x)^2 \|v_t\|^2 \tag{86}$$

$$+ \frac{12\Phi_*}{\gamma} \Big(1 + \sum_{t=1}^{T} (\|\nabla_x f(x_t, y_t; \xi_t^x)\|^2 + \|\nabla_y f(x_t, y_t; \xi_t^y)\|^2)\Big)^{1/3} + \frac{2}{7}\mathbb{E}\sum_{t=1}^{T}\|v_t\|^2,$$

Using Case 4 implies that

$$\frac{1}{2}\sum_{t=1}^{T} \|\nabla_x f(x_t, y_t)\|^2 + \sum_{t=1}^{T} \|\nabla_y f(x_t, y_t)\|^2$$

$$\leq 4\kappa L + \frac{3C_6}{2}\sum_{t=1}^{T}\left(\mathbb{E}\sum_{t=1}^{T} \left(\|v_t\|^2 + \|w_t\|^2\right)\right)^{1/3}$$

$$+ \frac{12\Phi_*}{\gamma}\Big(1 + 2\sigma^2 T + \sum_{t=1}^{T}(\|\nabla_x f(x_t, y_t)\|^2 + \|\nabla_y f(x_t, y_t)\|^2)\Big)^{1/3} \tag{87}$$

$$\leq 4\kappa L + \frac{3C_6}{2}\sum_{t=1}^{T}\left(2\mathbb{E}\sum_{t=1}^{T}\left(\|\nabla_x f(x_t, y_t)\|^2 + \|\nabla_y f(x_t, y_t)\|^2\right)\right)^{1/3}$$

$$+ \frac{12\Phi_*}{\gamma}\Big(1 + 2\sigma^2 T + \sum_{t=1}^{T}(\|\nabla_x f(x_t, y_t)\|^2 + \|\nabla_y f(x_t, y_t)\|^2)\Big)^{1/3}$$

It implies that:

$$\mathbb{E}\sum_{t=1}^{T}\|\nabla_x f(x_t, y_t)\|^2 + \mathbb{E}\sum_{t=1}^{T}\|\nabla_y f(x_t, y_t)\|^2 \leq O(T^{1/3}). \tag{88}$$

Conclude all the above cases, using Cauchy-Schwarz inequality, we can easily get

$$\frac{1}{T}\left[\mathbb{E}\sum_{t=1}^{T}\|\nabla_x f(x_t, y_t)\| + \mathbb{E}\sum_{t=1}^{T}\|\nabla_y f(x_t, y_t)\|\right]$$

$$\leq \frac{\sqrt{2}}{\sqrt{T}}\left[\sqrt{\mathbb{E}\sum_{t=1}^{T}\|\nabla_x f(x_t, y_t)\|^2 + \mathbb{E}\sum_{t=1}^{T}\|\nabla_y f(x_t, y_t)\|^2}\right] \leq O(\frac{1}{T^{1/3}}) \tag{89}$$

Then we finish the proof.

