# OpenReview forum: "VRAda: A Variance Reduced Adaptive Algorithm for Stochastic Parameter-Agnostic Minimax Optimizations"
_ICLR.cc/2024/Conference — ICLR 2024 Conference Withdrawn Submission_

### Official Review · Reviewer_QWNk · 2023-11-01

**Soundness:** 2 fair
**Presentation:** 2 fair
**Contribution:** 2 fair
**Rating:** 5
**Confidence:** 5

**Summary:**

This paper proposes an efficient variance-reduced adaptive algorithm (VRAda) to solve a class of nonconvex minimax optimization problems based on parameter-agnostic manner. Meanwhile, it provides the convergence analysis of the proposed method, and prove that it obtains an optimal sample complexity of $O(\epsilon^{-3})$ for finding an $ \epsilon$-stationary solution. It also provides some experimental results to verify the efficiency of the proposed method.

**Strengths:**

This paper proposes an efficient variance-reduced adaptive algorithm (VRAda) to solve a class of nonconvex minimax optimization problems based on parameter-agnostic manner. Meanwhile, it provides the convergence analysis of the proposed method, and prove that it obtains an optimal sample complexity of $O(\epsilon^{-3})$ for finding an $ \epsilon$-stationary solution. It also provides some experimental results to verify the efficiency of the proposed method.

**Weaknesses:**

This paper proposes an efficient parameter-agnostic adaptive algorithm for nonconvex-strongly-convex and nonconvex-PL minimax problems, however, the proposed method basically extends the existing STORM+ method to solve the minimax problems. Meanwhile, the proposed adaptive method requires a strict assumption of the bounded stochastic gradients (Given in Assumption 2).

**Questions:**

1)	In the proposed algorithm, the hyper-parameter $\beta_t$ is a key parameter. How to choose it in convergence analysis and experiments ?

2)	In the experiments, I think that the comparison methods are less. The authors should add some existing adaptive minimax optimization methods.

3)	The authors should detailly discuss the differences between the proposed method and the NeAda-AdaGrad method of [1], since the NeAda-AdaGrad method also is a parameter-agnostic adaptive algorithm for minimax optimization.

    [1] Nest Your Adaptive Algorithm for Parameter-Agnostic Nonconvex Minimax Optimization, NeurIPS 2022.

4)  Some typos, e.g., the equality (5) in the page 5.

---

### Official Review · Reviewer_byJD · 2023-11-02

**Soundness:** 1 poor
**Presentation:** 1 poor
**Contribution:** 2 fair
**Rating:** 3
**Confidence:** 4

**Summary:**

This paper aims to develop an adaptive (parameter agnostic) algorithm for solving stochastic minimax optimization when the problem is nonconvex-strongly concave or nonconvex-PL. The algorithm is using variance reduced estimator STORM of Cutkosky-Orabona and its "more adaptive" variant by Levy et al., 2021 to adapt to "varying inherent variance of primal and dual variables" which means in my understanding just using variance reduction. The claimed result is the oracle complexity of $O(1/\epsilon^{3})$ for finding points with small gradient norm w.r.t. $x$ and $y$. The authors claim that this result improves over the existing adaptive minimax algorithm TiAda of Li et al., 2022 and matches the best rate obtained with non-adaptive algorithms. Experiments are given at the end to compare with TiAda.

**Strengths:**

The goal of the paper, obtaining the optimal complexity guarantees for solving stochastic partially nonconvex min-max problems with parameters independent of problem constants such as Lipschitz constants and gradient norm upper bounds is interesting. Ideally, we would want algorithms to not require these constants and get the best complexity and of course, we want to have guarantees for nonconvex problems. Of course such a goal is easy to come up with, but difficult to implement correctly both in theory and practice. Moreover, three different problem instances are considered in the experiments which is also good. Consideration of different assumptions on the dual part (strong convexity or PL) is also useful since there is a lot of interest in PL and applications satisfying PL recently. This last point would be better if a trade-off would be given by the authors for these two different settings. Specifically, what do we gain additionally from assuming strong convexity rather than just PL.

**Weaknesses:**

Unfortunately, the paper is not satisfactory for many different considerations, namely: the correctness, the presentation, consistency between the claims and the obtained results, both in theory and practice. Below I only list evidence for some of them but there are many other examples of the same issues in the paper, which leads me to recommend a clear rejection. The paper needs to justify the proof steps and vastly improve writing and presentation.

- Many of the concepts that the authors are talking about are unclear. For example, I could count the phrase "varying inherent variances" used 4 times in the first 3 pages alone (2 times only in the abstract) without describing what the authors mean by this. Especially the authors say "varying inherent variances in variables $x$ and $y$" which is not clear at all. For example, one thinks that this might refer to different variance levels for $x$ and $y$, but as Assumption 3 shows this is not the case. It seems they only use this to mean that they use variance reduction whereas TiAda of the previous work does not. What is varying here? Please explain the phrases that you use, especially when they are not standard as "varying inherent variance".

- The first sentence of the abstract: it is a bit too far-fetched to claim that one line of research (parameter agnostic optimization with the language used in the paper) "bridges the gap between theoretical and empirical machine learning". This is very clearly false. The theoretical and empirical ML has so many other disconnects that are completely unrelated to parameter-agnostic optimization and reducing this to a single line of research is misleading. Such claims either should be supported by evidence or should be avoided since they lead to overselling of the results.

- Abstract also says the other approaches "require manually tuning problem-dependent parameters to attain an optimal solution". Again, a very imprecise phrase. Of course, none of the iterative algorithms used or referred in this paper attains an optimal solution. We always attain an approximately optimal solution. Next, it is not clear what this "manual tuning" refers to. For example, it is well known that SGD will have the same big-Oh complexity w.r.t. $\epsilon$ with any step size as a constant divided by $\sqrt{k}$ where $k$ is the number of iterations, the important part is the constant that can be very bad and this is what parameter-free optimization is trying to address, see for example https://arxiv.org/abs/2305.12475. However, this submission unfortunately has  vague claims that need to be clarified and made much more precise.

- Remark 1 states that "our proof does not necessitate the assumption of second order Lipschitz continuity for $y$, setting our proof on more rigorous compared to Li et al, (2022)" where Li et al, (2022) is the paper that introduced TiAda. Apart from the broken sentence (it should be "on more rigorous ground"), the sentence does not make much sense to me. Why would an assumption make a proof more or less rigorous? These are two different things that are being presented in a misleading way. The authors can say they have less assumptions and a stronger result, but to argue that they are "more rigorous" they have to show why the work of Li et al, (2022) is not rigorous. In fact, as I show below, I have questions about the rigor of this submissions since there are steps of the proof that are not justified (more on this below). Remark 4 also says obtaining a $O(\epsilon^{-3})$ complexity underscores the "rigor of our proof analysis". This is not what "rigor" means. A result cannot justify the rigor of the analysis, it is the other way around: a rigorous proof justifies a result.

Even though the above 2-3 points might look mostly a "style choice", they highlight a theme that goes on throughout the paper: the paper has many unclear notions, claims and the paper has at times misleading way of presenting the results leading one to suspect the contribution is more than what is shown in the paper.

Now I move on to other important topics, correctness:

- The proof of Lemma 5, beginning of page 22: Here, the authors have an inner product that they want to bound in expectation: $<w_t - \nabla_y f(x_t, y_t; \xi_t^y), {y_t}^* - y_t>$. Of course, when one uses expectation conditioned on $y_t$, then one can take the expectation w.r.t. $\xi_t^y$, but the authors are doing something more here, they unroll the expectation in eq. (34) and take expectations w.r.t $\xi_{t-1}^y$ and so on to show that the "total" expectation of the first argument of the inner product is 0 and they conclude that the inner product has expectation $0$. This is definitely not justifiable since $\xi_{t-1}^y$ is not anymore independent of $y_t$ and hence this unrolling of the expectation couples the arguments of this inner product and the argument is not correct. Such a recursive argument as in the paper should be made directly on the inner product, not just one argument. The authors need to fix this proof.

I did not go much more further in checking the proofs since this step is giving already pretty low confidence for the correctness. Just while a quick reading, other issues are also present:

- Moving on to the proof of Lemma 9, third line of eq. (69) has incorrect constants (Young's inequality will bring addiional factor of $2$), which is not very important. But then in eq. (70), one of the error terms is just missing. What happens to the $L/2 \| y_{t+1} -y_t\|^2$ that is in eq. (68), why does it not appear in eq. (70)?

Related works:

- Sect 2: Maybe I missed, but I could not see the paper mention any other work (parameter agnostic or otherwise) that obtains $O(\epsilon^{-3})$, why? Because Remark 2 claims that there are indeed "parametric algorithms" (which in my understanding means the algorithms that are not "parameter agnostic") that achieve this complexity. Please describe them clearly in your related works section.

Experiments:

- Even though the goal of the paper is to have a parameter agnostic algorithm, experiments use tuning for parameters such as $\beta_t$ which we learn in Appendix A.2. This unfortunately makes the practical results not very convincing since the main goal of the algorithm is to avoid tuning. To make a case for the algorithm in the paper, one needs to have experiments with no tuning. Because it is possible that more tuning on the baselines can lead to better results than the algorithms proposed in this submission.

Smaller points that make the presentation more problematic:

- After Assumption 3, the paper says "It is worth noting that these assumptions are only presented to facilitate our proof. In the implementation of VRAda, we do not need any information from them to achieve the final result" which is misleading and also not correct. Of course, with any optimization algorithm, we need the assumptions to prove something, not to implement the algorithm. But we *do* need information from the assumptions in practice too: we need to know that the assumptions hold in our empirical problem, for our theory to be applicable. Otherwise, the algorithm becomes a heuristic. In summary, such a phrasing is problematic and unnecessary. It is problematic because it undermines the importance of assumptions and the connection between theoretical assumptions and practical problems. Especially, since the first sentence of the abstract states that the paper's ambitious aim is to "bridge the gap between theoretical and empirical machine learning", I think the importance of assumptions should be highlighted, compared to what is done now.

- The way citations are written is quite unreadable. This is because the references do not have any paranthesis or brackets. I believe this is because the authors used \cite instead of \citep, please correct this. At many places, we have text such as (taken from the end of page 1) "like AdaGrad Duchi et al. (2011), AMSGrad Reddi et al. (2018), and STORM+ Levy et al. (2021)",  no paranthesis, exactly as written here. Hence, it is not clear at all what is part of the citation and what is not, one can understand as long as they know that AdaGrad, AMSGrad and STORM+ are algorithms and Duchi, Reddi and Levy are last names, but of course this kind of citing continues throughout the paper where it is sometimes less clear and makes it very difficult to read.

- Before eq. (79), QG Growth should be just QG since this way it is "quadratic growth growth".

- Please organize the proofs better. Lemma 3 has a 4 page proof and Theorem 1 has a 6 page proof. These proofs can be easily organized to split to sub-results. Also, one can organize the proofs much better with clearer steps in between. Please reconsider improving the writing with the reader in mind.

**Questions:**

The section "Weaknesses" already has many questions. Some questions below are also summarizing the points there:

- How is the proof of Lemma 5 justifiable since unrolling the expectation in eq. (34) couples the two arguments of the inner product? Such a recursive argument as in the paper should be made directly on the inner product, not just one argument.

- What happens to the $L/2 \| y_{t+1} -y_t\|^2$ that is in eq. (68), why does it not appear in eq. (70)?

- Why is there no comparison with Huang, 2023 "Enhanced adaptive algorithms for nonconvex-pl minimax optimization"?

- Why is there a need for having $\mathcal{Y}$ as the domain for the dual variable? Why can't one just take the whole space? This is quite problematic since they assume right after Assumption 3 that dual solution is in the interior which is a quite artificial assumption. Why is this needed?

- beginning of page 5: "gradient calculations of the two sub-problems". What is the subproblem here? The algorithm does not solve any subproblems. These are not the gradients of sub-problems, they are gradients of functions.

- what do we gain additionally from assuming strong convexity rather than just PL?

---

### Official Review · Reviewer_TWBa · 2023-11-03

**Soundness:** 2 fair
**Presentation:** 3 good
**Contribution:** 2 fair
**Rating:** 3
**Confidence:** 3

**Summary:**

The paper presents a variance reduction algorithm for nonconvex-strongly-concave (NC-SC) and nonconvex-Polyak-Łojasiewicz (NC-PL) minimax optimization that eliminates the need for manual hyperparameter tuning or prior knowledge of problem parameters. Specifically, for both scenarios, it achieves a convergence rate of $O(T^{1/3})$ for seeking a near-stationary point.

**Strengths:**

- This work represents the first study on a variance reduction algorithm for these settings that successfully circumvents the necessity for hyper-parameter tuning.
- The paper is well-organized, offering a clear and logical structure that facilitates comprehension.

**Weaknesses:**

- **Novelty.** The idea of the paper is not too surprising, since it combines the idea of STORM [Cutkosky & Orabona, 2019] and TiAda [Li et al, 2022]. Despite the anticipated synergy of these ideas, the paper demonstrates an effort in the careful design and analysis required for such a combination.
- **Domain constraint for y.** The paper addresses scenarios where the variable $y$ is confined within the domain $ \mathcal{Y} $. Given the assumptions that the stochastic gradient is bounded and the function exhibits strong convexity with respect to $ y $, it follows that the domain $ \mathcal{Y} $ must be finite in diameter. However, the absence of any projection in the algorithm raises concerns, as the iterates for $ y $ could significantly deviate from the domain, leading to the generation of irrelevant points. Although the paper sidesteps certain technical challenges by assuming the gradient at the optimal solution is zero, the lack of projection is still a notable issue.
- **Writing.** The clarity of the writing could be improved. Certain phrases are ambiguous and could benefit from further elaboration. For example,  the discussion at the end of page 3 regarding “unresolved issue of varying inherent variance of….” and “faces difficulties adapting to the optimal trajectory”.

**Questions:**

1. In Remark 3, the paper mentioned the other paper TiAda needs another two parameters $\alpha$ and $\beta$. It seems that TiAda needs different stepsize scale in $t$ in order to adapt to the time scale, which is shown to be necessary in an example mentioned in [Yang et al., 2022]:
                          $f(x, y) =  -\frac{1}{2} y^2 + Lxy - \frac{L^2}{2}x^2.$
       I wonder why this paper does not need different scales on $t$ in the stepsizes for $x$ and $y$. Although I was not able to check each step in the proof, I suspect that the proposed algorithm may not converge for the aforementioned example with no noise:
Considering the example provided, where $\nabla_x f(x, y) = Ly - L^2x$ and $\nabla_y f(x, y) = -y + Lx$, it follows that $\nabla_x f(x, y) = -L \nabla_y f(x, y)$. Consequently, given the definitions of $v_t$ and $w_t$, we can deduce that $v_t = -Lw_t$ for all t. Then by the definition of $\alpha_t^x$ and $\alpha_t^y$, we have $\alpha_t^x = L^2 \alpha_t^y$. It further leads to $\eta_t^x = \eta_t^y/L^{2/3}$, since $\alpha\_t^x > \alpha\_t^y$ with $L>1$ in the definition of  $\eta\_t^x$ and $\eta\_t^y$. Now in the update of $x$ and $y$, we have $-\eta_t^x v_t = (\eta_t^y w_t)\cdot L^{1/3}$. Therefore, if $ x_0 = y_0 \cdot L^{1/3} $, it follows that $ x_t = y_t \cdot L^{1/3} $, which suggests non-convergence to the stationary point unless it approaches $(0,0)$. However, given the strong concavity in $ x $, convergence to $(0,0)$ from some initializations seems improbable.
The authors have demonstrated the algorithm’s performance on an alternative toy function in their experiments. It would be informative to know if they have also tested the algorithm on the function described above.

2.  I wonder why the paper needs to present results for NC-SC and NC-PL at the same time, given that NC-PL setting has already included NC-SC setting.

---

### Official Review · Reviewer_TzGV · 2023-11-07

**Soundness:** 2 fair
**Presentation:** 3 good
**Contribution:** 2 fair
**Rating:** 3
**Confidence:** 4

**Summary:**

The paper introduces VRAda, a parameter-agnostic, variance-reduced adaptive algorithm. It’s designed for tackling the challenge of varying inherent variances in stochastic minimax optimizations without relying on problem-specific parameters.

**Strengths:**

1. **Problem Setting and parameters:** VRAda is parameter-agnostic, and specifically, tailored for Non-Convex-Strongly-Concave (NC-SC) and Non-Convex-Polyak-Łojasiewicz (NC-PL) minimax optimization scenarios.

2. **Theoretical:** The paper offers a theoretical analysis showing that VRAda, under specific assumptions, can achieve a sample complexity of $\mathcal{O}(1/\varepsilon^3)$, for finding an \\( \varepsilon \\)-stationary point in NC-SC and NC-PL settings.

3. **Empirical Performance:** When compared empirically to existing algorithms such as TiAda, VRAda demonstrates better performance in identifying \\( \varepsilon \\)-stationary points.

**Weaknesses:**

1. **Assumption Constraints:** The assumptions (1, 2, and 3) appear to be very strong and counter-productive to the main goals of the paper. Assuming bounded gradients and bounded variances, which is at odds with tasks where variance is typically high and/or unbounded — *the very scenarios where variance reduction is most crucial.* This restricts the theory and real-world applicability of the work.

2. **Algorithm Stability:** Stability is a core concern, as the algorithm's momentum scale (\\(\beta\\)) and learning rate (\\(\eta\\)) are derived from the inverse sum of gradient norms. In cases where gradients exhibit significant variance could introduce instability, leading to unpredictable updates and challenging convergence. A robust approach should ideally handle mechanisms for broader range of gradient behaviors, particularly in the tails of the distribution.

3. **Experimental Validation:** The experiments conducted do not fully substantiate the theoretical and practical benefits of VRAda. For instance, functions ($f(x, y) = log(1 + x^2) − xy + y^2/2$ and $f(x,y) = −x^3+xy−y^2$) are easily solvable analytically and do not adequately challenge the algorithm. Meanwhile, a limited GAN experiment — focusing only on loss curves — fail to rigorously test the algorithm's capabilities or provide insight into the quality of the generated samples.

4. **Theoretical Bounds:** The presented bounds on expected values in Theorems 1 and 2 are less persuasive than high-probability bounds, which would better represent the full distribution of outcomes. For stochastic problems, it's essential to quantify the likelihood and extent of deviations from the mean. This is particularly relevant if the underlying distributions are not tightly concentrated around their mean — a common scenario in practice.

5. **Sample Complexity Claim:** Remark 2 regarding sample complexity is a direct outcome of the initial assumptions and a specific equation (17). The general applicability of this claim without these assumptions is not explored, nor is the handling of larger batch sizes, which is mentioned in the introduction but not adequately addressed later on.

**Questions:**

**Theorems (1, 2):** Can the authors clarify the sources of randomness in the expectations, and assess whether the results extend to high-probability bounds? This would provide more comprehensive understanding of VRAda's performance in various conditions.


**On Assumptions (1,2,3) and practical relevance:** Could the authors explore and discuss VRAda's main results in scenarios where the strong assumptions (1, 2, 3) are relaxed?

**On Stability with Variable Gradients:** How does gradient in the tails of the distribution impact the stability and performance of VRAda? Could the paper provide an analysis of this aspect?

**On Experimental Validation**: Would it be possible to include more complex and varied experimental tasks to showcase the strengths and limitations of VRAda? Alternatively, a more detailed analysis of the results, like qualitative assessments in applications such as GANs?